# LLaMP: Large Language Model Made Powerful for High-fidelity Materials Knowledge Retrieval

## Abstract

Reducing hallucination of Large Language Models (LLMs) is imperative for use in the sciences, where reliability and reproducibility are crucial. However, LLMs inherently lack long-term memory, making it a nontrivial, *ad hoc*, and often biased task to fine-tune them on domain-specific literature and data. Here we introduce **LLaMP**, a multimodal retrieval-augmented generation (RAG) framework of hierarchical reasoning-and-acting (ReAct) agents that can dynamically and recursively interact with computational and experimental data from the Materials Project (MP) and run atomistic simulations via high-throughput workflow interface. Without fine-tuning, LLaMP demonstrates strong tool-usage ability to comprehend and integrate various modalities of materials science concepts, fetch relevant data stores on the fly, process higher-order data (such as crystal structure and elastic tensor), and streamline complex tasks in computational materials and chemistry. We propose a metric combining uncertainty and confidence estimates to evaluate the self-consistency of responses by LLaMP and vanilla LLMs. Our benchmark shows that LLaMP effectively mitigates the intrinsic bias in LLMs, counteracting the errors on bulk moduli, electronic bandgaps, and formation energies that seem to derive from mixed data sources. We also demonstrate LLaMP's capability to edit crystal structures and run annealing molecular dynamics simulations using pre-trained machine-learning interatomic potentials. The framework offers an intuitive and nearly hallucination-free approach to exploring and scaling materials informatics, and paves the way for future agentic scientific workflows and knowledge-grounded LLMs.

## 1 Introduction

The generation of convincing yet unreliable information poses a pressing challenge to large language model (LLMs), particularly to their application in the sciences. LLMs are prone to hallucination–providing outright false information with high confidence (Bang et al., 2023; Xu et al., 2024). This issue is particularly concerning for knowledge-intensive tasks, where users rely on chatbots and other AI systems to provide accurate guidance (Lewis et al., 2020). LLMs often lack up-to-date factual knowledge on topics outside their training data, requiring rigorous verification against trusted external sources (Mallen et al., 2023). In the scientific community, where the integration of insights and data accuracy is already complex, the proliferation of generative models may exacerbate the risk of misinformation. This trend accentuates the importance of scrutinizing and ensuring the reliability of information sources.

Current approaches to enhance LLM accuracy in domain-specific knowledge often involve fine-tuning pre-trained models (Dagdelen et al., 2024; Gupta et al., 2022) or tailored prompt engineering techniques (Yang et al., 2023; Zheng et al., 2023). While these models are easy to deploy, they suffer from diminished reproducibility and data adherence due to the absence of a memory base, untraceable fine-tuning history, or opaque extraction processes. Even though fine-tuning can encode a certain amount of domain-specific knowledge into LLMs, it is constrained by scalability and intrinsic memory capacity. Fine-tuned LLMs struggle to retain in the long term the knowledge they were trained on as the training progresses, nor can they be aware of the recent events and data beyond pre-training. Prompt engineering, while effective, also compromises the generalizability, thus limiting

the overall power and flexibility of LLMs. Therefore, a more sensible approach involves equipping LLMs with external data sources, allowing them to generate holistic responses via few-shot adaptation to factual information (Lewis et al., 2021) that can reliably support real-world scientific research and decision-making.

In this work, we propose LLaMP, a multimodal retrieval-augmented generation (RAG) framework leveraging hierarchical reasoning-and-acting (ReAct) agents to interact with Materials Project (MP), arXiv, Wikipedia, and atomistic simulation tools. The framework serves as a safeguard against LLM hallucination by grounding them in high-fidelity material informatics from large-scale material database~~derived from various sources~~, including ~~computational data from~~ quantum-mechanical first-principles calculations and expert-curated material synthesis recipes, and further enables the capabilities of complex downstream tasks. The hierarchical planning of supervisor and assistant ReAct agents improves self-correcting tool-usage performance and enhances the self-consistency in final responses. The new capabilities emerge—such as multi-modal searching, tensor and 3D crystal structure retrieval and operation, and language-driven simulation. The framework~~Through hierarchical planning of multiple ReAct agents, we demonstrate that LLaMP~~ not only can correctly retrieve high-fidelity, higher-order materials data~~higher-order materials data such as tensors and 3D crystal structures~~ but also can combine different modalities to perform complex, knowledge-intensive inferences and operations essential for real-world materials science applications.

Our contributions are as follows: (1) we introduce a multimodal RAG framework employing hierarchical ReAct agents that dynamically interact with the Materials Project, enabling LLMs to access high-fidelity materials informatics; (2) we propose a statistical metric to assess the self-consistency of LLM responses in high-precision, reproducibility-critical settings; (3) we evaluate the performance of LLaMP and standard LLMs in predicting key material properties, including bulk moduli, electronic bandgaps, formation energies, and magnetic orderings; (4) we showcase real-world applications in materials science, such as inorganic synthesis and crystal structure generation and editing; (5) we enhance LLaMP with high-throughput atomistic simulation workflows and pre-trained universal ML force fields, lowering the entry barriers to computational materials and chemistry.

## 2 BACKGROUND

**Materials Project (MP)** The Materials Project is a multi-institution effort to explore and compute the properties of all known inorganic materials (Jain et al., 2013) and molecules (Spotte-Smith et al., 2023). The initiative leverages high-throughput electronic structure calculations (Kresse and Furthmüller, 1996; Shao et al., 2015) based on density functional theory (DFT), providing large-scale open-source database and analysis algorithms, with the ultimate goal to drastically reduce the time and cost required for materials discovery by focusing experiments on the promising candidates from computational screening. Most of the atomic structures are selected from the Inorganic Crystal Structure Database (ICSD) (Zagorac et al., 2019) and undergo standardized relaxation procedures, followed by post-processing or additional calculations for higher-order material properties such as electron and phonon bandgaps, elastic tensors, dielectric tensors, and more. MP provides these calculated material properties through API endpoints.

**NLP and LLM in materials science** Natural language processing (NLP) has found extensive application in extracting valuable information from scientific publications, with notable instances involving text-to-text or more recent image-to-text summarization techniques (Gupta et al., 2022; Radford et al., 2021; Tshitoyan et al., 2019). For summarizing crystal structures in textual form, Ganose and Jain (2019) introduced the *robocrystallographer*, a toolkit designed for the analysis and generation of descriptions for crystalline materials. Their method condenses atomic structures into descriptive JSON representations that encompass coordination statistics, connectivity motifs, geometric features, and dimensionality. MP leverages robocrystallographer to generate human-level descriptions for 130K compounds which are accessible through MP website and API.

Recent efforts have curated datasets (Zaki et al., 2023) and benchmarks (Song et al., 2023) to better evaluate the limitations of LLMs in question answering within the materials science domain. Zhang et al. (2024) further curated instruction data to fine-tune Llama for material science-specific tasks. These works focus on general (undergraduate-level) question answering instead of factual grounding on expert-curated database and downstream agentic workflow. In a complementary aspect, other

works address the challenges of extracting complex materials informatics from diverse formats such as tables and unstructured texts (Hira et al., 2024; Schilling-Wilhelmi et al., 2024). This motivates us to augment LLM's knowledge base with MP—one of the most authorative materials database of stable crystal structures, high-fidelity DFT calculations, inorganic solid-state synthesis recipes, *etc.*

## 3 RELATED WORK

**Prompting and fine-tuning in domain science**  Prompt-based methods have been used as effective tools for automating data extraction process from the literature. Polak and Morgan (2023) employ a prompt workflow to extract the cooling rates of metallic glasses and yield strengths of high entropy alloys. Zheng et al. (2023) implement a ChatGPT metal-organic framework (MOF) synthesis assistant through embedding and searching on preselected papers. StructChem (Ouyang et al., 2024) leverages step-by-step reasoning, and iteratively refines results to solve college-level chemistry questions. Yang et al. (2023) use GPT-4 to extract experimentally measured bandgaps to train a graph neural network for accurate bandgap prediction from crystal structures. Despite the success in the specific data extraction tasks, prompt-based methods face challenges in reproducibility when the used prompts are fine-grained to work for specific edge cases. They are also still prone to hallucination and less generalizable to combine different data sources due to the deliberately designed prompt.

Several other knowledge-grounded, domain-specific language models lean on the fine-tuning approach against pre-selected data and literature. For instance, ChemGPT (Frey et al., 2022) involves fine-tuning GPT-neo on self-referencing embedded strings (SELFIES) representations of small molecules. Jablonka et al. (2024) demonstrated GPT-3 fine-tuned against online corpora could outperform purpose-trained models on classification, regression, and inverse design of high-entropy alloys and molecules. Dagdelen et al. (2024) fine-tuned GPT-3 on ∼500 prompt-completion pairs to enhance LLM's capability to extract useful information on materials chemistry from text paragraphs. However, the fine-tuned models without augmentation inherently lack awareness of the up-to-date results and any data only available after their training. Moreover, fine-tuned LLMs still suffer from limited memory retention and are prone to forget during continual training (Wang et al., 2023).

**LLM function calling and tool usage**  An emerging class of LLM applications, including this work, take advantage of LLM text completion and instruction following capability for function calling. This approach extends LLMs with expert-curated tools to improve the quality of control for downstream applications. Coscientist (Boiko et al., 2023) combines tools such as search engines, Python, and document index for autonomous chemical research. ChemCrow (M. Bran et al., 2024) gathers multiple molecule and safety tools to enhance organic chemistry experiment and molecule design. Concurrently, Zhang et al. (2024) develop retrieval based agentic framework on their curated dataset. Ghafarollahi and Buehler (2024) propose AtomAgents for alloy design and analysis.

However, most prior works adopt *flat planning* strategy, where a single agent accesses all the available tools, resulting in a lack of self-correcting tool usage capabilities. This often leads to premature reasoning stop and summarization when the agent encounters tool usage errors. We mitigate this through *hierarchical planning* of multiple ReAct agents (see Section 4.1).

## 4 METHOD

### 4.1 HIERARCHICAL ORCHESTRATION

**Overviews**  Flat planning, where an agent see all the available tools and related API schemas, quickly exceeds LLM context window and incurs huge cost for large-scale database like MP. To manage heterogeneous data sources and diverse types of queries, we introduce hierarchical planning, featuring a supervisor ReAct agent overseeing multiple assistant ReAct agents that have access to the tools (Figure 1). This design offers three major advantages over flat planning commonly implemented in previous works (Boiko et al., 2023; M. Bran et al., 2024): (1) modularity of the system ensures that each assistant agent can focus on domain-specific queries while the supervisor agent handles higher-level reasoning and task allocation; (2) the hierarchical structure improves

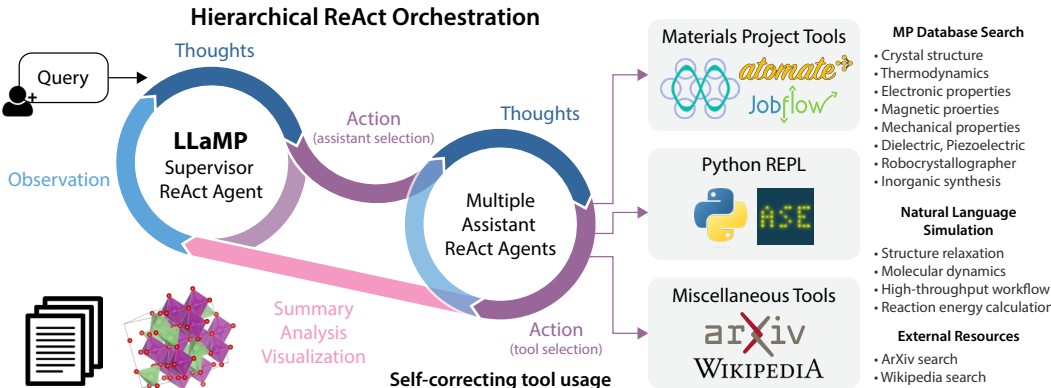

Figure 1: Hierarchical ReAct agent planning in LLaMP. Two levels of agents are deployed using a standardized LangChain interface (Chase, 2022). Supervisor ReAct agent oversees assistant ReAct agents at the bottom-level, each equipped with distinct toolkits and data/document stores to accomplish various tasks, including high-fidelity materials information retrieval, atomistic modeling and simulations, and literature search. For a detailed example, refer to Figure A.1.

the overall accuracy and efficiency by reducing the cognitive load on any individual agent; (3) by offloading specific functions to specialized agents, we minimize the context window consumption and schema parsing.

**Supervisor agent** The supervisor agent acts as a router and decision-maker, handling abstract logic between user requests and assistant agents. Here, we adopt ReAct on GPT-4 (Yao et al., 2023) to augment the agent's action space $\mathcal{A}$ with a language space $\mathcal{L}$ to create an expanded action space of $\hat{\mathcal{A}} = \mathcal{A} \cup \mathcal{L}$. This expanded action space empowers the agent to take action $\hat{a}_t \in \mathcal{L}$ in language space that facilitate the collaboration with assistant agents to retrieve domain specific information and achieve complex downstream tasks such as molecular dynamics simulations.

**Assistant agent** The efficient function calling in LLMs is often hindered by the need to process complex API schemas, which can consume a significant portion of the context window. To address this, we assign a specialized ReAct agent for each specific tool or API endpoint. It reduces context window consumption, as each agent handles only the relevant schema for its task, avoiding unnecessary schema parsing. Additionally, the use of ReAct agents enables them to refine their API calls based on feedback, significantly improving task completion rates through ReAct's iterative self-correcting mechanism.

The full list of agents and tools are defined in A.1. Each MP assistant agent employs a self-correcting ReAct mechanism, enabling agents to refine their API calls and improve task completion rates. The framework's modularity enable a seamless integration of new assistant agents, allowing for extensibility to various materials discovery methods and experimental techniques (Luo et al., 2023; Pilania et al., 2017; Wen et al., 2023; 2024; Zeni et al., 2024).

## 4.2 SELF-CONSISTENCY OF RESPONSE (SCoR)

When LLMs are integrated in scientific workflows and deployed in high-stakes settings (*i.e.* self-driving labs), it is important for these models to have consistent and predictable behaviors (Liang et al., 2023). For numeric knowledge retrieval tasks, we define the following metrics:

**Precision** (sample standard deviation) measures the uncertainty in the model's responses where $n$ is the number valid responses from $N$ trials and $\hat{\sigma}$ is the standard deviation of valid response:

$$\text{Precision} = \frac{\hat{\sigma}}{\sqrt{n}} \geq 0$$

.

**Coefficient of Precision (CoP)** maps the precision to $(0, 1]$:

$$\text{CoP} = \exp\left(-\text{Precision}\right) = \exp\left(-\frac{\hat{\sigma}}{\sqrt{n}}\right) \in (0, 1].$$

**Confidence** measures the ratio of generating $n$ valid responses in $N$ trials:

$$\text{Confidence} = \frac{n}{N}.$$

**Self-consistency of Response (SCoR)** is then defined as

$$\text{SCoR} = \text{CoP} \times \text{Confidence} \in [0, 1].$$

The limit of $\text{SCoR} = 1$ is reached when the model yields the same response to a given query every time. At the limit of $\text{SCoR} = 0$, the model is either very inconsistent (with large variance across the responses) or very reluctant (with low confidence) to answer the query. Despite the simplicity in definition, SCoR effectively reflects the reproducibility and practical usability of the method, which is important when the method is incorporated into broader systems where the stable and expected behaviors are prioritized. Refer to Appendix A.2 for the detailed procedure of metric calculation.

## 5 EXPERIMENTS

### 5.1 MULTIMODAL REACT AUGMENTATION

Materials design often involves multi-objective property optimization. These properties span a Pareto front where optimizing one factor incurs deterioration in others. To succeed in such tasks, combining different modalities of materials properties is necessary. LLaMP achieves this through the hierarchical orchestration of multiple ReAct agents (Yao et al., 2023). For the example question "*What's the stiffest material with the lowest formation energy in Si-O system?*" (Figure A.1), when a query requires multimodal information and compound logic, the supervisor agent decomposes the query into multiple subtasks, delegates them to assistant agents (MPThermoExpert and MPElasticityExpert) for information retrieval, and in the final stage of reasoning integrates information from both modalities, drawing on the context in episodic memory retrieved from the assistant agents (Figure 1). This enables LLaMP to achieve various tasks step-by-step by combining multiple data sources from the Materials Project (MP) (*e.g.* 3D crystal structures, thermodynamic, mechanical, magnetic properties, and more listed in Appendix A.1) in a single query.

### 5.2 PERFORMANCE BENCHMARKS

**Response quality and consistency** We evaluate the performance of LLaMP, StructChem (Ouyang et al., 2024), Darwin (Xie et al., 2023), and vanilla LLMs (gpt-4, llama3-8b, gemini-1.0-pro) on material properties such as bulk modulus, formation energy, and bandgap (Figure 2, Table 1). Performance is assessed through Precision, CoP, SCoR, and MAE metrics, as defined in Section 4.2. We argue that any useful LLM agents to be included in the scientific workflow should have high SCoR and low error on the materials properties. Notably, LLaMP consistently outperforms other models, achieving the highest SCoR and the lowest errors across material properties, making it highly suitable for scientific workflows. StructChem, despite extensive prompting strategies, often fails due to a lack of necessary domain knowledge, resulting in high refusal rates when it cannot validate outputs.

For bulk modulus prediction, vanilla LLMs, particularly Llama 3-8b, frequently rely on low-fidelity online data, leading to significant deviations for elements like Cr, Mn, and Fe, compared to MP theoretical values. Interestingly, Llama 3-8b usually cites spurious reference in the responses despite largest response variance but occasionally agrees with MP values. In contrast, LLaMP outperforms vanilla LLMs and reduces the MAE from around 40 to 14.57 GPa.

Our results demonstrate that vanilla LLMs fail to provide accurate formation energy predictions, with low SCoR and high MAE ranging from 1.5 to 5.5 eV, which is impractical for material discovery requiring $\text{meV}$-level precision. This is not unexpected, since accurate formation energy prediction requires the computation of multiple energetics (energies of the compound itself and its elemental constituents).

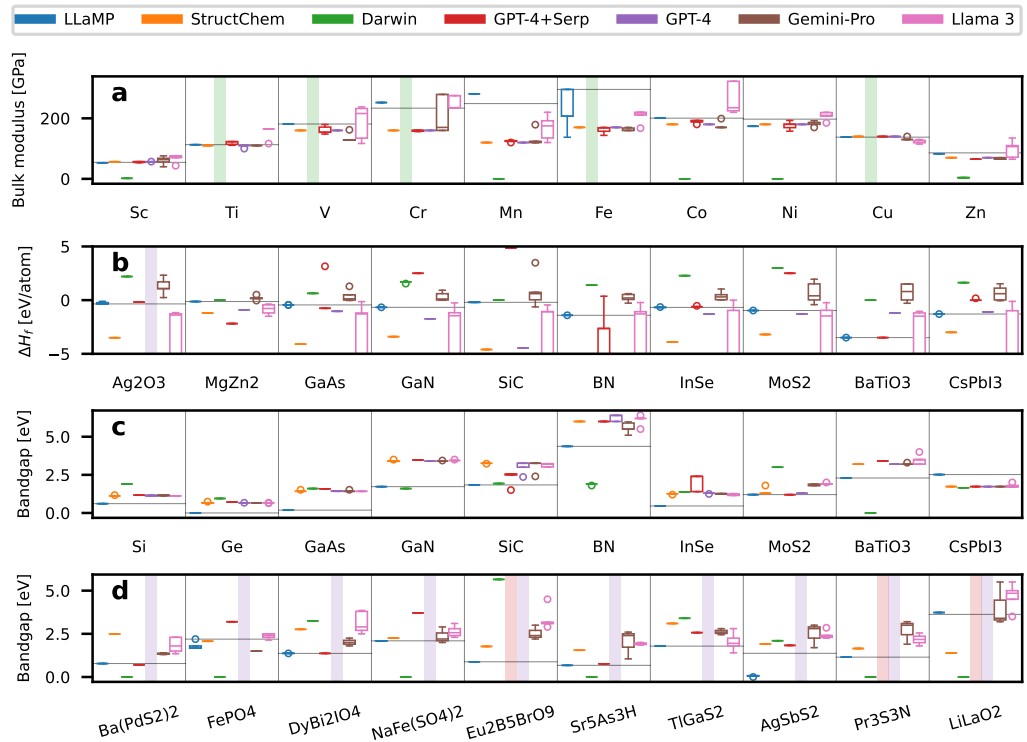

Figure 2: Boxplot of LLaMP RAG responses, baseline methods, and LLM intrinsic knowledge on material properties. (a) Bulk moduli, $K$, of 3d transition metals. (b) Formation energies, $\Delta H_f$, of common compounds. (c) Electronic bandgaps, $E_g$, of common intrinsic semiconductors. (d) Electronic bandgaps of multi-element (ternary or quaternary) materials. Missing predictions are marked by shaded areas. Fliers (Outliers) are marked in circles. Horizontal lines represent the MP reference data. All LLaMP results use GPT-4 as backend language provider. Method with higher SCoR has narrower distribution. LLaMP is effectively grounded on MP reference across different tasks and materials.

In evaluating bandgaps, we query 10 common compounds and 10 multi-element materials that are less commonly encountered in the literature. Vanilla LLMs perform surprisingly well on the bandgaps of common semiconductors (Figure 2c), with expected systematic deviation from MP values retrieved by LLaMP[1]. This is likely due to the extensive literature on experimental semiconductor bandgaps, which have been studied and reported for decades. On the contrary, vanilla LLMs lack intrinsic knowledge of the bandgaps for the queried multi-element materials and exhibit low confidence or refuse to make predictions (Figure 2d, Table B6.8), whereas LLaMP retrieves accurate data with a SCoR of 0.938 and correctly identifies the stable polymorph's bandgap when multiple forms are present.

**Ablation study** Our frameworks relies on two principal components: first, factual material informatics on MP database; second, stable function calling mechanism that allows assistant agent to interact with tools. In Table 5, we examine three variants: (1) LLaMP: ReAct with MP tools; (2) GPT-4+ReAct with SerpAPI for internet browsing; (3) vanilla GPT-4. LLaMP achieved the best performance when using the complete set of MP tools, highlighting the importance of grounding in up-to-date, high-fidelity materials databases. In Section 4.1, we mentioned the importance of hierarchical planning for robust function call. Evaluating several backbone models on bulk moduli and formation energy prediction, we found LLaMP's grounding performance correlates with the

---

[1]Bandgaps calculated from generalized gradient approximation (GGA) functional are known to underestimate the experimental values by 40-50% (Borlido et al., 2020). Strategies to improve bandgap prediction at moderate or low computational cost will be included in MP in the future.

Table 1: Performance metrics of LLaMP and LLM baselines on the prediction of material properties. The metrics from left to right are precision (sample standard deviation), coefficient of precision (CoP), confidence, self-consistency of response (SCoR), and mean absolute error (MAE), where Materials Project are taken as the ground truth. All the tabulated values are the average metrics over five runs and the sampled materials. All LLaMP and StructChem results use GPT-4 as backend language provider. Better method has high SCoR and MAE simultaneously.

| | Bulk Modulus $K$ (GPa) | | | | | Formation Energy $\Delta H_f$ (eV) | | | | |
|---|---|---|---|---|---|---|---|---|---|---|
| | Precision↓ | CoP | Confidence | SCoR↑ | MAE↓ | Precision↓ | CoP | Confidence | SCoR↑ | MAE↓ |
| LLaMP (GPT-4) | 2.698 | 0.900 | 1.000 | 0.900 | **14.574** | 0.006 | 0.994 | 0.940 | 0.934 | 0.007 |
| LLaMP (Sonnet) | 1.816 | 0.562 | 1.000 | 0.562 | 15.104 | 0.000 | 1.000 | 1.000 | **1.000** | **0.000** |
| LLaMP (Gemini) | 5.178 | 0.053 | 1.000 | 0.053 | 16.251 | 0.076 | 0.932 | 0.620 | 0.576 | 0.166 |
| LLaMP (Llama3) | 12.993 | 0.036 | 0.800 | 0.029 | 50.308 | 0.000 | 1.000 | 0.250 | 0.250 | 1.377 |
| StructChem | 0.000 | 1.000 | 0.200 | 0.200 | 41.017 | 0.000 | 1.000 | 0.200 | 0.200 | 3.146 |
| Darwin | 0.001 | 0.999 | 0.500 | 0.499 | 156.266 | 0.003 | 0.997 | 1.000 | 0.997 | 2.245 |
| GPT-4+Serp | 2.221 | 0.833 | 0.300 | 0.433 | 29.937 | 0.025 | 0.977 | 0.560 | 0.791 | 11.669 |
| GPT-4 | 0.186 | 0.910 | 1.000 | 0.910 | 41.225 | 0.000 | 1.000 | 0.180 | 0.200 | 1.680 |
| Sonnet | 0.009 | 0.992 | 1.000 | **0.992** | 41.033 | 0.022 | 0.979 | 1.000 | 0.979 | 294.360 |
| Gemini-Pro | 6.065 | 0.169 | 1.000 | 0.169 | 43.429 | 0.467 | 0.657 | 1.000 | 0.657 | 1.412 |
| Llama 3 | 11.222 | 0.010 | 1.000 | 0.010 | 41.874 | 2.346 | 0.139 | 0.960 | 0.137 | 4.657 |

| | Electronic Bandgap $E_g$ - Common (eV) | | | | | Electronic Bandgap $E_g$ - Multi-element (eV) | | | | |
|---|---|---|---|---|---|---|---|---|---|---|
| | Precision↓ | CoP | Confidence | SCoR↑ | MAE↓ | Precision↓ | CoP | Confidence | SCoR↑ | MAE↓ |
| LLaMP (GPT-4) | 0.000 | 1.000 | 0.800 | 0.800 | **0.000** | 0.047 | 0.958 | 0.960 | 0.918 | **0.167** |
| LLaMP (Sonnet) | 0.145 | 0.870 | 0.600 | 0.522 | 0.298 | 0.046 | 0.962 | 1.000 | 0.962 | 0.304 |
| LLaMP (Gemini) | 0.627 | 0.571 | 0.600 | 0.343 | 1.327 | 0.003 | 0.997 | 0.500 | 0.997 | 0.637 |
| LLaMP (Llama3) | 0.051 | 0.952 | 0.800 | 0.761 | 1.038 | 0.169 | 0.848 | 0.800 | 0.678 | 1.094 |
| StructChem | 0.017 | 0.984 | 1.000 | 0.984 | 0.986 | 0.000 | 1.000 | 0.200 | 0.200 | 0.973 |
| Darwin | 0.002 | 0.998 | 1.000 | 0.998 | 1.224 | 0.000 | 1.000 | 1.000 | **1.000** | 1.951 |
| GPT-4+Serp | 0.040 | 0.963 | 1.000 | 0.963 | 1.012 | 0.000 | 1.000 | 0.660 | 0.660 | 0.576 |
| GPT-4 | 0.032 | 0.970 | 1.000 | 0.970 | 0.959 | - | - | 0.000 | 0.000 | - |
| Sonnet | 0.000 | 1.000 | 1.000 | **1.000** | 0.938 | 0.000 | 1.000 | 0.500 | 1.000 | 0.644 |
| Gemini-Pro | 0.034 | 0.968 | 1.000 | 0.968 | 0.994 | 0.168 | 0.849 | 0.600 | 0.509 | 0.989 |
| Llama 3 | 0.042 | 0.960 | 1.000 | 0.960 | 1.053 | 0.182 | 0.836 | 0.860 | 0.719 | 1.091 |

function-calling capability of backbone LLM: Claude-3.5-Sonnet (#1) > Gemini-1.5-Flash (#24) > and Llama3-8B (#46). The number following each model refers to its ranking on the Berkeley Function-Calling Leaderboard at the time of the experiment (Yan et al., 2024).

**High-fidelity and higher-order data retrieval** The challenge for LLMs in excelling at knowledge- and data-intensive tasks is well-documented (Cobbe et al., 2021; Hendrycks et al., 2021; Liang et al., 2023). Figure 3 shows the prediction of LLaMP, GPT-3.5, and GPT-4 on the magnetic orderings and total magnetization of 800 materials randomly selected from all unary, binary, and ternary compounds in MP. Our result indicates that without RAG, vanilla LLMs suffer from hallucinations and misclassify the magnetic orderings of materials. LLaMP with GPT-4 as backend can counteract the intrinsic bias of GPT models, increasing the classification accuracy to 0.98 and $R^2$ of magnetization prediction to 0.992 (Table 2). We note that GPT-3.5 as backend, while effective for classification and other information retrieval tasks, struggles to distinguish `total_magnetization` from `magnetization_per_formula_unit` in magnetism API schema and often requests the wrong field and forgets to normalize the values. In the magnetic orderings queries, LLaMP with GPT-3.5 as backend fails to distinguish ferromagnetic (FM) and ferrimagnetic (FiM) orderings, while LLaMP with GPT-4 as backend gracefully separates the two classes (Figure 3a, d).

We further test the capability of LLaMP and LLMs for higher-order data (such as tensors, 3D crystal structures, curves). As shown in Table B6.2, GPT-3.5 hallucinates the values for the components in the elastic tensor of NaCl, with serious erroneous values such as $C_{11} = 289.2$ GPa—a significant deviation from DFT-calculated values (76 GPa). It also omits the values for $C_{22}, C_{33}, C_{55}, C_{66}$ and fails to represent the full elastic tensor in a matrix format, despite the query explicitly requesting the *full* elastic tensor. This hightlights the limitation of intrsinic knowledge in LLMs to recall higher-

Table 2: Prediction performance of LLaMP, GPT-3.5, and GPT-4 on magnetic orderings and magnetization. LLaMP with GPT-4 and GPT-3.5 as backend LLM are compared.

| | Magnetic Ordering | | Magnetization | |
|---|---|---|---|---|
| | Accuracy | F1 | MAE | $R^2$ |
| LLaMP (GPT-4) | **0.98** | **0.89** | **0.045** | **0.992** |
| GPT-4 | 0.48 | 0.26 | 1.611 | -0.201 |
| LLaMP (GPT-3.5) | 0.96 | 0.88 | 1.896 | 0.407 |
| GPT-3.5 | 0.23 | 0.18 | 1.988 | -0.024 |

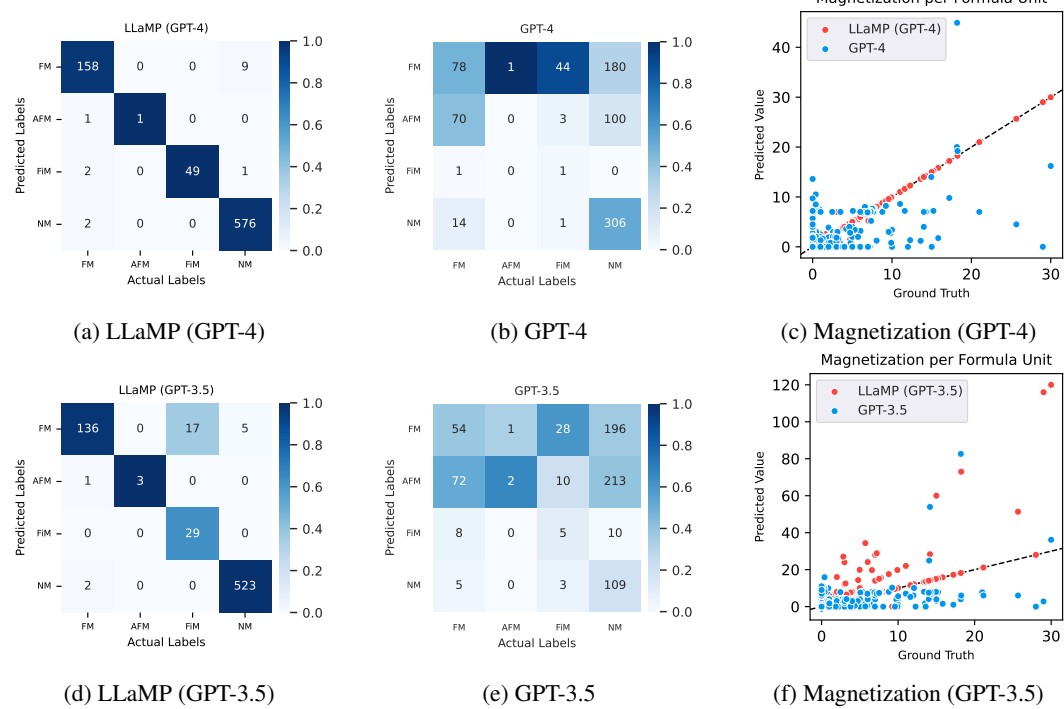

Figure 3: Prediction of LLaMP, GPT-3.5, and GPT-4 on (a,b,d,e) magnetic orderings and (c,f) total magnetization per formula unit of randomly selected materials. Confusion matrix presents the number of entries in each class. Colormap represents the percentage of correct classification.

order, more complex data for more comprehensive, holistic response.

## 5.3 REAL-WORLD APPLICATIONS

**Inorganic synthesis recipes** Empowered by the MP synthesis endpoint (Kononova et al., 2019), LLaMP can extract synthesis recipes and summarize detailed step-by-step procedures grounded on real experimental papers with associated DOI references, as demonstrated in the example queries (Table B6.9 and B6.10).

Vanilla LLMs often give seemingly correct and verbose synthesis procedures but pull irrelevant compounds into the recipes and overlook more optimal or efficient reactions. In the example of $YMnO_3$ (Table B6.9), GPT-3.5 suggests the possible reaction pathways from two common oxide precursors ($Y_2O_3$ and $MnO_2$). However, it pulls irrelevant lithium compounds ($Li_2CO_3$ and $LiOH$) into the recipe and overlooks the fact that metathesis reactions (Li et al., 2015; Todd et al., 2021) require less applied energy than high-temperature sintering, which relies on solid-state diffusion (Maximenko and Olevsky, 2004).

Table 3: Positive-unlabeled (PU) classification of LLaMP and baseline methods on inorganic material synthesizablity. (*) Evaluations on 352,236 positive and 40,817 unlabeled compounds by Kim et al. (2024).

|  | Accuracy | F1 | Precision | Recall |
|---|---|---|---|---|
| LLaMP (GPT-4) | 0.800 | 0.773 | **0.895** | 0.680 |
| LLaMP (Sonnet) | **0.818** | **0.812** | 0.848 | 0.780 |
| GPT-4 | 0.600 | 0.649 | 0.578 | 0.740 |
| Sonnet | 0.530 | 0.230 | 0.636 | 0.140 |
| Llama3 | 0.480 | 0.623 | 0.489 | **0.860** |
| Gemini | 0.590 | 0.388 | 0.765 | 0.260 |
| GPT-4* | - | - | 0.151 | 0.620 |
| GPT-3.5 (FT)* | - | - | **0.558** | **0.951** |
| stoi-CGNF* | - | - | 0.541 | 0.942 |

Vanilla LLMs also exhibit uncertainty about specific synthesis details, such as heating temperature, duration, cooling rate, *etc*. In some edge cases such as $LiFePO_4$ presented in Table B6.10, the cited

references are associated with the real papers but the contents are dissociated from the cited titel and hallucinated from the pre-training corpus.

We further compare the performance of LLaMP on synthesizability prediction with stoichiometric convolutional graph neural fingerprint (stoi-CGNF) (Jang et al., 2024) and fine-tuned LLMs (Kim et al., 2024). We follow the positive-unlabeled (PU) classification task proposed in (Kim et al., 2024) by randomly selecting a subset of positive (probable) and unlabeled (unlikely) inorganic compounds and compare the classification performances of LLaMP with different backend LLMs and baselines. As presented in Table 3, LLaMP effectively enhances the performances of backbone GPT-4 and Sonnet LLMs by a significant margin of 20%, with the classification precision of LLaMP (GPT-4) up to 0.895.

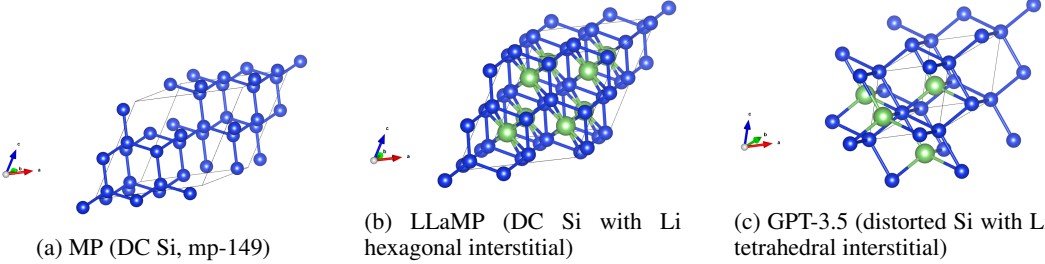

(a) MP (DC Si, mp-149)
(b) LLaMP (DC Si with Li hexagonal interstitial)
(c) GPT-3.5 (distorted Si with Li tetrahedral interstitial)

Figure 4: Generation and manipulation of crystal structures using LLMs to insert an additional lithium atom at the interstitial site in diamond cubic silicon structure. Blue: Si. Green: Li. Question-answer pairs are listed in Table B6.11. Additional atoms extended through bonds are visualized.

Table 4: Structural parameters of the generated crystals compared with diamond cubic (DC) silicon. From left to right are fractional coordinates of inserted Li atom $(x, y, z)_{\text{Li}}$, total cell volume $V$, average Si−Si bond lengths $\ell_{\text{SiSi}}$, Si−Si−Si angles $\theta_{\text{SiSiSi}}$, and Si−Li−Si angles $\theta_{\text{SiLiSi}}$. GPT-4 refuses to respond due to their safeguard against the lack of atomic structure information.

| | $(x, y, z)_{\text{Li}}$ | $\ell_{\text{SiSi}}$ (Å) | Error (%) | $V$ (Å$^3$) | Error (%) | $\theta_{\text{SiSiSi}}$ (°) | Error (%) | $\theta_{\text{SiLiSi}}$ (°) |
|---|---|---|---|---|---|---|---|---|
| LLaMP | $(0.5, 0.5, 0.5)$ | 2.36 | **0.0** | 40.33 | **0.0** | 109.47 | **0.0** | 62.96 |
| GPT-3.5 | $(0.5, 0.5, 0.5)$ | 2.71 | +15.0 | 67.05 | +66.3 | 98.28 | -10.2 | 67.69 |
| GPT-4 | - | - | - | - | - | - | - | - |
| DC Si (mp-149) | | 2.36 | | 40.33 | | 109.47 | | |

**RAG-assisted crystal generation and editing** Fine-tuned LLMs for text-encoded atomistic information have shown the capability to generate stable crystals under the constraints of atomic positions and charges (Gruver et al., 2023). In this context, we delve into the examination and comparison of the crystal generation capabilities between LLaMP and GPT-3.5, without resorting to fine-tuning or tailored prompt messages in previous work. Figure 4 showcases the structures generated by LLaMP and vanilla GPT-3.5 without RAG, both instructed to *insert one lithium atom at the tetrahedral site of the diamond cubic silicon structure* (Table B6.11). Notably, both LLaMP and GPT-3.5 place an additional Li atom at fractional coordinate $(0.5, 0.5, 0.5)$. However, the Si structure retrieved by LLaMP adheres to the MP convention, positioning two Si bases at $(0.125, 0.125, 0.125)$ and $(0.875, 0.875, 0.875)$. This causes the inserted Li atom to be *hexagonal interstitial* instead of *tetrahedral interstitial*.

GPT-3.5 locates the Li atom at the tetrahedral site given the "luckily chosen" Si bases at $(0, 0, 0)$ and $(0.25, 0.25, 0.25)$; however, the resulting cell volume and shape are highly distorted, and the Si−Si bond length and Si−Si−Si angle deviate significantly from the ground truth (Table 4), highlighting the limitations in the intrinsic encoding of LLMs for atomistic information and the challenges associated with zero-shot generation of crystal structures. In contrast, the LLaMP-retrieved MP structure serves as a robust prior, anchoring the lattice parameters of the generated structure to the correct values.

**Language-driven simulation**  LLaMP equipped with Python REPL and atomistic simulation workflow package `atomate2` performs well out of the box for complex multi-step simulations using pre-trained universal machine learning interatomic potential MACE-MP-0 (Batatia et al., 2023) through language instruction. As demonstrated in Appendix C.1 and Appendix C.2, LLaMP is able to follow multi-step instruction to fetch stable crystal structure from MP, generate a supercell of atomic structure, and run annealing molecular dynamics simulation with varying temperature from 300K to 800K and back to 300K. After the simulation is finished, LLaMP can read the simulation trajectories and plot the temperature profile over time (Appendix C.1).

We further test the robustness of our language-driven workflow on running MD simulations (Figure 5). A subset of 50 supercell structures were randomly created from up-to-quinary compounds in MP. Each MD simulation runs $0.1\,\mathrm{ps}$ with timestep of $2\,\mathrm{fs}$. The timeout was set to 90 seconds. 96% workflows (SUCCESS+TIMEOUT) were successfully initiated, with 62% finished and 34% of systems exceeding 90 seconds timeout due to slow or stalled MACE-MP-0 runs (the simulation is still running without error but runs slowly). 4% simulations ran into unspecified status (UNKNOWN). We found that during these triggered workflows LLaMP asks user for approval on the precise chemical formula to fetch the structure from MP, rendering the workflow unfinished.

## 6  DISCUSSION

**Robustness**  The hierarchical ReAct framework implemented here is essentially a graph of agents, or *language graph*, with one central node (supervisor) in connection with many satellite nodes (assistants). The implementation of ReAct for the assistant agents enables self-correcting tool usages and fortifies the robustness of data retrieval. As presented in Figure A.1c, MPThermoExpert initially misunderstood the schema at the first trial and filled in the formula field with $Si-O$, an invalid input but a valid one for chemical system (chemsys) field. The observation step (step 4) allows MPThermoExpert to handle exceptions and to refine the correct input fields after adaptation (step 6). Storing (Retrieving) question-answer and query-argument pairs to (from) vector databases could further reduce the number of trial-and-error steps, and the stored pairs can be used to refine foundation LLMs to improve function calling quality.

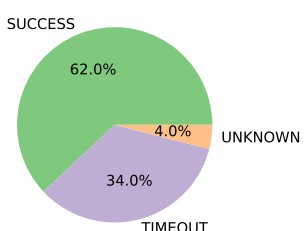

Figure 5:  The final four statuses (SUCCESS, TIMEOUT, FAILURE, UNKNOWN) of trial language-driven MD simulation runs on random MP supercell structures. LLaMP successfully initiated 96% (SUCCESS+TIMEOUT) of all the simulations within 90 second timeout window.

**Limitation**  We recognize the effectiveness of LLaMP's framework relies on backbone LLM's function calling and reasoning capabilities. Sometimes LLMs misunderstand the description of schemas and therefore yield unexpected behaviors. For example, `sort_fields` argument allows sorting the returned documents in ascending order or descending order if the field is prefixed with $-$, but LLMs sometimes mistake the sign and sort in the opposite order. Other example failure or safeguard modes are presented in Appendix A.3. The correctness of LLaMP is also subject to the quality of theoretical prediction and the comprehensiveness of the data in MP. Other than the underpredicted bandgaps by GGA functional, MP's ongoing effort to search all possible magnetic configurations is also not complete. Most of the existing calculations in MP start from high-spin ferromagnetic configurations, which may overlook many antiferromagnetic ground states below the current energy convex hull. While MP is one of the most comprehensive materials databases, the available crystal structures on MP are not exhaustive but continuously expanding (Merchant et al., 2023), and would be benefited from additional intermetallic compounds and high-entropy materials from other databases such as AFLOW, OQMD, NOMAD, *etc.* (Curtarolo et al., 2012; Kirklin et al., 2015; Scheidgen et al., 2023). Furthermore, Kohn-Sham DFT theory is insufficient in some cases, and a higher level of theory is needed. Currently LLaMP only supports a few `atomate2` workflows with machine learning force fields and VASP calculations. More diverse electronic calculation methods and workflows will be supported in the future work.

**Summary** We present a hierarchical agentic framework, LLaMP, based on ReAct to extract and manipulate material informatics through few-shot generalization. By grounding thoughts and actions with high-fidelity information, LLaMP showcases the ability to integrate various modalities of material properties and perform logical inferences to accomplish assigned tasks, all without the need for fine-tuning. In essence, the proposed LangChain framework holds the potential to expand its applicability to multiple data sources, encompassing both theoretical computations and experimental data, knowledge graph databases (Venugopal and Olivetti, 2024; Ye et al., 2024), and real-world laboratories by incorporating additional assistant agents for data retrieval and robot control (Fei et al., 2024). LLaMP functions as a knowledge-aware agent, empowering users to navigate and manipulate complex materials informatics. In the context of self-driving labs (Boiko et al., 2023; Szymanski et al., 2023), LLM agents with multimodal data sources, sensors, and actors may improve their decision making and operation (Miret and Krishnan, 2024). As new tools continue to emerge, there is an exciting avenue for further exploration to ascertain if this framework can effectively facilitate scientific hypothesis generation and guide data-driven experiments.

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

# A  SUPPLEMENTARY INFORMATION

## A.1  LIST OF IMPLEMENTED ASSISTANT AGENTS AND TOOLS

Here we provide the comprehensive list of implemented **assistant agents** and `tools`. Note that **MP Assistants** are highly modular so it is very trivial to support extra API endpoints from https://api.materialsproject.org/docs.

- **MPSummaryExpert**: `summary` provides amalgamated data for a material by combining subsets of data from many of the other API endpoints.

- **MPThermoExpert**: `thermo` provides computed thermodynamic data for a material such as formation energy and energy above hull.

- **MPElasticityExpert**: `elasticity` provides bulk, shear, and Young's modulus, poisson ratio, and universal anisotropy index.

- **MPMagnetismExpert**: `magnetism` provides computed magnetic ordering related data.

- **MPDielectricExpert**: `dielectric` provides computed dielectric data from density functional perturbation theory.

- **MPPiezoelectricExpert**: `piezoelectric` provides computed piezoelectric data from density functional perturbation theory.

- **MPElectronicExpert**: `electronic_structure` provides computed electronic structure related data for a material such as band gap and fermi level. Python objects for line-mode band structures, density of states, and fermi surfaces are also available.

- **MPSynthesisExpert**: `synthesis` provides a synthesis recipes for materials extracted from literature using text mining and natural language processing techniques.

- **MPStructureRetriever**: `MaterialsStructureText` fetches and saves pymatgen Structure objects to local JSON files.

- **MLFFAgent**: `MLFFMD` runs molecular dynamics simulations using pre-trained machine learning force fields; `MLFFElastic` calculates the elastic constants of a given material using pre-trained machine learning force fields.

- `PythonREPLTool`: Python REPL that LLMs could run the generated script.

- `ArxivQueryRun`: LangChain built-in tool that LLMs can use to send API request to ArXiv.

- `WikipediaQueryRun`: LangChain built-in tool that LLMs can use to send API reqeust to Wikipedia.

## A.2  METRIC CALCULATION IN TABLE 1 AND FIGURE 2

The following procedures are adopted to calculate the metrics for material property regression benchmarks presented in Table 1 and Figure 2:

1. Each method was presented with the same query asking for the property of multiple materials. Here we ask each method for bulk modulus, formation energy, and electronic bandgap of ten materials. For example, "What are the bulk moduli of the following metals: Sc, Ti, V, Cr, Mn, Fe, Co, Ni, Cu, Zn?".

2. Repeat the same query for five times and collect the responses.

3. The numerical values are extracted and formatted into 2D arrays with the aid by LLMs. If the range is provided in the response (*e.g.* Llama 3), the median value was used.

4. Calculate Precision, CoP, Confidence, and SCoR for each method across five trials on all materials. The code is provided and the pseudocode can be written as follows:

---
**Input:** `arr` (2D array of five responses for different materials)
$n \leftarrow$ count of valid responses (non-NaN values) in each column of `arr`
$\texttt{prec} \leftarrow \frac{\texttt{nanstd(arr,axis=0)}}{\sqrt{n}}$
$\texttt{cop} \leftarrow \text{mean}(\exp(-\texttt{prec}))$
$\texttt{conf} \leftarrow \text{mean}\left(\frac{n}{\texttt{\# of trials } N}\right)$
$\texttt{scor} \leftarrow \begin{cases} 0 & \text{if } \texttt{conf} = 0 \text{ for all columns} \\ \texttt{cop} \times \texttt{conf} & \text{otherwise} \end{cases}$
$\texttt{prec} \leftarrow \text{mean}(\texttt{prec})$
**Output:** $\texttt{prec}, \texttt{cop}, \texttt{conf}, \texttt{scor}$

---

### A.3 SAFEGUARD OR FAILURE MODES

- **Question: What is the bulk modulus of stainless steel?** `ambiguous chemical formula`
  **LLaMP:** Stainless steel is an alloy with a composition that can vary widely, including elements like iron, chromium, nickel, and molybdenum. Because of this variability, it does not have a single chemical formula or a unique Materials Project ID, and its bulk modulus can significantly vary based on its exact composition and heat treatment. The Materials Project database, focusing on inorganic crystalline materials, may not provide specific data on complex alloys like stainless steel. For precise values of the bulk modulus, consulting specific datasheets or literature on the type of stainless steel in question is recommended.

- **Question: How to synthesize Y2Be2Fe(SiO5)2** `data unavailable`
  **LLaMP:** Currently, there are no synthesis recipes available in the Materials Project database for Y2Be2Fe(SiO5)2.

- **Question: What's the phonon bandgap of Ac2HgGe** `computed property unavailable`
  **LLaMP:** The material Ac2HgGe, with a Materials Project ID of mp-861872, does not have its phonon bandgap information directly available through the Materials Project database. For detailed phonon properties, further analysis using specialized phonon calculation tools or databases would be necessary.

Table 5: Performance of LLaMP with different backbone LLMs and ablation of ReAct agent with MP API and SerpAPI tools.

| | Bulk Moduli $K$ (GPa) | | | | | Formation Energies $\Delta H_f$ (eV) | | | | |
|---|---|---|---|---|---|---|---|---|---|---|
| | Precision↓ | CoP | Confidence | SCoR↑ | MAE↓ | Precision↓ | CoP | Confidence | SCoR↑ | MAE↓ |
| LLaMP | 2.698 | 0.900 | 1.000 | **0.900** | **14.574** | 0.007 | 0.993 | 0.960 | 0.953 | 0.009 |
| LLaMP (Sonnet) | 1.345 | 0.702 | 0.867 | 0.608 | 16.121 | 0.000 | 1.000 | 1.000 | **1.000** | **0.000** |
| LLaMP (Gemini-1.5) | 21.586 | 0.217 | 1.000 | 0.217 | 64.324 | 0.390 | 0.718 | 0.560 | 0.402 | 0.531 |
| LLaMP (Llama 3) | 7.314 | 0.327 | 0.800 | 0.261 | 47.386 | 0.413 | 0.721 | 1.000 | 0.721 | 3.062 |
| GPT-4+Serp | 2.222 | 0.352 | 1.000 | 0.352 | 41.742 | 5.947 | 0.745 | 1.000 | 0.745 | 8.214 |
| GPT-4 | 0.186 | 0.910 | 1.000 | 0.910 | 41.225 | 0.000 | 1.000 | 0.180 | 0.180 | 1.680 |

**a. User Query**

What's the stiffest material with the lowest formation energy in Si-O system?

**b. LLaMP ReAct (Supervisor)**

*1-Thought:* To answer this question, I need to find materials in the Si-O system with the lowest formation energy and the highest stiffness. I will use the MPThermoExpert tool to search for materials in the Si-O system and sort them by formation energy. After finding candidates, I will need to use the MPElasticityExpert tool to determine the stiffness of these materials.

*2-Action:*
```json
{
  "action": "MPThermoExpert",
  "action_input": {
    "input": "What are the materials with the lowest formation energy in the Si-O system?"
  }
}
```

*9-Thought:*
The previous work has identified materials in the Si-O system with the lowest formation energy. The next step is to determine the stiffness of these materials. I will use the MPElasticityExpert tool to find the stiffness (typically represented by the bulk modulus or Young's modulus) for each of these materials.

*10-Action:*
```json
{
  "action": "MPElasticityExpert",
  "action_input": {
    "input": "What are the bulk and Young's moduli for Si2O5 (mp-862998), SiO2 (mp-733790), SiO2 (mp-6922), SiO2 (mp-556985), and SiO2 (mp-556994)?"
  }
}
```

*14-Thought:*
The output from the MPElasticityExpert tool provides bulk modulus values for the Si-O materials with the lowest formation energy. However, it seems there was a mismatch between the material IDs requested and the material IDs returned in the output. The Young's modulus values are not provided, but the bulk modulus can be a good indicator of stiffness. The material with the highest bulk modulus would be considered the stiffest.

From the provided output, the SiO2 with material ID mp-9258 has the highest bulk modulus of 310.27 GPa, which suggests it is the stiffest among the listed materials. Since this material also belongs to the Si-O system and has a high bulk modulus, it is likely to be one of the stiffest materials with low formation energy in the Si-O system.

*15-Answer:*
The stiffest material in the Si-O system with low formation energy is likely SiO2 with material ID mp-9258, which has a bulk modulus of 310.27 GPa. The Young's modulus values were not available, but the bulk modulus is a strong indicator of stiffness.

**c. Specialized ReAct (Assistant)**

**MPThermoExpert**

*3-Action:*
```json
{
  "action": "search_materiasls_thermo__get",
  "action_input": {
    "formula": "Si-O",
    "sort_fields": "formation_energy_per_atom",
    "limit": 5,
    "fields": "material_id,formula_pretty,formation_energy_per_atom"
  }
}
```

*4-Observation:*
Error on search_materials_thermo__get: REST query returned with error status code 400 on URL https://api.materialsproject.org/materials/thermo/?_fields=material_id... Problem processing one or more provided formulas.

*5-Thought:*
The query may be too broad or ambiguous. I will refine the search by specifying the chemical system more precisely.

*6-Action:*
```json
{
  "action": "search_materials_thermo__get",
  "action_input": {
    "chemsys": "Si-O",
    "sort_fields": "formation_energy_per_atom",
    "limit": 5,
    "fields": "material_id,formula_pretty,formation_energy_per_atom"
  }
}
```

*7-Observation:*
[{'formula_pretty': 'Si2O5', 'material_id': 'mp-862998', 'formation_energy_per_atom': -2.289431383571429}, {'formula_pretty': 'SiO2', 'material_id': 'mp-733790', 'formation_energy_per_atom': -3.039244523055555}, ...]

*8-Answer:*
The materials with the lowest formation energy in the Si-O system are: Si2O5 (mp-862998) with a formation energy per atom of -2.289 eV/atom, SiO2 (mp-733790) with a formation energy per atom of -3.039 eV/atom, SiO2 (mp-6922) with a formation energy per atom of -3.037 eV/atom, SiO2 (mp-556985) with a formation energy per atom of -2.993 eV/atom, and SiO2 (mp-556994) with a formation energy per atom of -2.958 eV/atom.

**d. Specialized ReAct (Assistant)**

**MPElasticityExpert**

*11-Action:*
```json
{
  "action": "search_materials_elasticity__get",
  "action_input": {
    "formula": "Si2O5,SiO2,SiO2,SiO2,SiO2",
    "fields": "material_id,formula_pretty,bulk_modulus,young_modulus"
  }
}
```

*12-Observation:*
[{'formula_pretty': 'SiO2', 'material_id': 'mp-1021503', 'elastic_tensor': {'raw': [[355.5072134787483, 236.22725456386155, 236.22725456386155, 2.081668171172168e-16, 7.771561172376094e-16, 7.771561172376094e-16], ...]

*13-Answer:*
The bulk and Young's moduli for the requested materials are as follows: Si2O5 (mp-1021503) - Bulk modulus: 275.987 GPa, Young's modulus: Not available; SiO2 (mp-11684) - Bulk modulus: 110.695 GPa, Young's modulus: Not available; SiO2 (mp-2374583) - Bulk modulus: 89.352 GPa, ...

Figure A.1: Multimodal retrieval-augmented generation for materials informatics. (a) User query. (b) Supervisor ReAct agent capable of handling multiple assistant agents and high-level reasoning. (c-d) Assistant ReAct agents executing function calling and summarization. (c) MPThermoExpert and (d) MPElasticityExpert have access to the API schemas of thermo and elasticity endpoints on Materials Project, respectively. The selected details are highlighted in red, demonstrating the capabilities of RAG and ReAct implemented in LLaMP. The blue texts show LLaMP assistant ReAct agent can handle API calling errors and self-correct the input query accordingly.

## A.4 PROMPT TEMPLATE

We use the ReAct template hwchase17/react-multi-input-json from LangChain Hub (https://smith.langchain.com/hub/hwchase17/react-json) as follows:

```
Answer the following questions as best you can. You have access to the following
    tools:

{tools}

The way you use the tools is by specifying a JSON blob.
Specifically, this JSON should have an 'action' key (with the name of the tool to
    use) and an 'action_input' key (with the input to the tool going here).

The only values that should be in the "action" field are: {tool_names}

The $JSON_BLOB should only contain a SINGLE action, do NOT return a list of multiple
    actions. Here is an example of a valid $JSON_BLOB:

'''
{{
  "action": $TOOL_NAME,
  "action_input": $INPUT
}}
'''

ALWAYS use the following format:

Question: the input question you must answer
Thought: you should always think about what to do
Action:
'''
$JSON_BLOB
'''
Observation: the result of the action
... (this Thought/Action/Observation can repeat N times)
Thought: I now know the final answer
Final Answer: the final answer to the original input question

Begin! Reminder to always use the exact characters 'Final Answer' when responding.
```

The above system prompt message is prepended with the following solicitation to guide the supervisor agent:

```
You name is LLaMP and you are a helpful agent that can consult materials-related
    data through Materials Project (MP) database, arXiv, Wikipedia, and run
    calculations/simulations via a python REPL and MP atomate2 computational
    workflow.

If you get an error, debug your code and try again. Only use the output of your code
    to answer the question. Ask user to clarify their queries if needed. Please
    note that you don't have direct control over MP but through multiple assistant
    agents to help you. You need to provide complete context in the input for
    assistants to do their job. REFINE and CRITQUE the output of the assistants if
    needed. You can end the conversation by saying "Bye".
```

# B  SAMPLE QUESTIONS

Table 6: QA comparison between LLaMP and GPT-3.5.

|  | Question | Answer | |
|---|---|---|---|
|  |  | **LLaMP** | **GPT-3.5** |
| B6.1 | What are the bulk moduli of the following metals: Sc, Ti, V, Cr, Mn, Fe, Co, Ni, Cu, Zn? | The bulk moduli of the metals you asked for are as follows (in GPa): - Scandium (Sc): Voigt = 45.715, Reuss = 45.34, VRH = 45.528 - Titanium (Ti): Voigt = 112.955, Reuss = 112.807, VRH = 112.881 - Vanadium (V): Voigt = 181.232, Reuss = 181.232, VRH = 181.232 - Chromium (Cr): Voigt = 252.155, Reuss = 252.155, VRH = 252.155 - Manganese (Mn): Voigt = 280.959, Reuss = 280.959, VRH = 280.959 - Iron (Fe): Voigt = 296.003, Reuss = 295.207, VRH = 295.605 - Cobalt (Co): Voigt = 200.587, Reuss = 200.587, VRH = 200.587 - Nickel (Ni): Voigt = 197.399, Reuss = 197.29, VRH = 197.345 - Copper (Cu): Voigt = 151.394, Reuss = 151.394, VRH = 151.394 - Zinc (Zn): Voigt = 76.283, Reuss = 95.46, VRH = 85.872 | The bulk modulus (K) is a measure of a material's resistance to uniform compression. It is defined as the ratio of the change in pressure to the fractional volume compression. The values for the bulk modulus can vary somewhat depending on the specific conditions and crystallographic structures of the materials. The values provided here are approximate and given in gigapascals (GPa): 
 - Scandium (Sc): 55 GPa - Titanium (Ti): 110 GPa - Vanadium (V): 160 GPa - Chromium (Cr): 160 GPa - Manganese (Mn): 120 GPa - Iron (Fe): 170 GPa - Cobalt (Co): 180 GPa - Nickel (Ni): 180 GPa - Copper (Cu): 140 GPa - Zinc (Zn): 70 GPa 
 These values represent typical bulk moduli for these metals, and actual values may vary depending on factors such as temperature and pressure. |

| | Question | Answer | |
|---|---|---|---|
| | | **LLaMP** | **GPT-3.5** |
| B6.2 | What is the full elastic tensor of NaCl? | The elastic tensor for NaCl is available for two different entries on Materials Project. For material˙id 'mp-22851', the elastic tensor in IEEE format is [[76.0, 1.0, 1.0, -0.0, 0.0, -0.0], [1.0, 76.0, 1.0, -0.0, -0.0, 0.0], [1.0, 1.0, 76.0, 0.0, -0.0, -0.0], [-0.0, -0.0, 0.0, -3.0, -0.0, -0.0], [0.0, -0.0, -0.0, -0.0, -3.0, 0.0], [-0.0, 0.0, -0.0, -0.0, 0.0, -3.0]]. For material˙id 'mp-22862', the elastic tensor in IEEE format is [[47.0, 12.0, 12.0, 0.0, -0.0, -0.0], [12.0, 47.0, 12.0, 0.0, -0.0, -0.0], [12.0, 12.0, 47.0, 0.0, 0.0, -0.0], [0.0, 0.0, 0.0, 12.0, -0.0, -0.0], [-0.0, -0.0, 0.0, -0.0, 12.0, 0.0], [-0.0, -0.0, -0.0, -0.0, 0.0, 12.0]] | The elastic tensor describes the relationship between stress and strain in a material. It is a 4th-rank tensor that characterizes the elastic properties of a crystal. In the case of NaCl (sodium chloride), the elastic tensor is typically represented by a 6x6 matrix. The elastic constants of NaCl can vary slightly depending on the specific conditions and crystallographic orientation. Here is an approximate form of the elastic tensor for NaCl: $C11 = 289.2$ GPa $C12 = 97.8$ GPa $C44 = 81.6$ GPa $C13 = C14 = C15 = C16 = C23 = C24 = C25 = C26 = 0$ GPa Note: The values are given in gigapascals (GPa). The elastic constants Cij represent the components of the elastic tensor. In this case, C11, C12, and C44 are the non-zero elastic constants. The off-diagonal terms C13, C14, C15, C16, C23, C24, C25, and C26 are typically zero due to cubic symmetry. Keep in mind that these values are approximate, and there may be slight variations depending on the source and the specific conditions under which the measurements were made. Experimental techniques, such as Brillouin scattering or ultrasonic measurements, are often used to determine the elastic constants of materials. |
| B6.3 | What are the bandgaps of the following materials: Si, Ge, and GaAs, GaN, SiC, BN, InSe, MoS2, BaTiO3, CsPbI3? Only look for ground-state bandgaps. | The ground state bandgaps for the requested materials are as follows: Si: 1.12 eV, Ge: 0.66 eV, GaAs: 1.42 eV, GaN: 3.28 eV, SiC: 2.36 eV, BN: 6.4 eV, InSe: 1.26 eV, MoS2: 1.29 eV, BaTiO3: 3.2 eV, CsPbI3: 1.73 eV. Please note that these values are typical for bulk materials and can vary depending on the specific structure and calculation method. | As of my last knowledge update in January 2022, I can provide you with approximate values for the ground state bandgaps of the materials you mentioned. Keep in mind that these values can vary depending on factors like temperature and crystal structure. Additionally, more recent research may have refined these values. Here are the approximate ground state bandgaps in electron volts (eV): Silicon (Si): 1.1 eV Germanium (Ge): 0.67 eV Gallium Arsenide (GaAs): 1.42 eV Gallium Nitride (GaN): 3.4 eV Silicon Carbide (SiC): 3.2 eV (for 4H-SiC) Boron Nitride (BN): 6 eV Indium Selenide (InSe): 1.25 eV Molybdenum Disulfide (MoS2): 1.3 eV (indirect bandgap) Barium Titanate (BaTiO3): 3.2 eV Cesium Lead Iodide (CsPbI3): 1.7 eV Please note that these values are general approximations and may not represent the specific conditions or variations in different forms of these materials. For the most accurate and up-to-date information, I recommend checking recent scientific literature or databases. |

Table 6 – continued from previous page

| | Question | Answer | |
| | | **LLaMP** | **GPT-3.5** |
|---|---|---|---|
| B6.4 | Give me the decomposition enthalpies of the following materials: Ag2O3, MgZn2, GaAs, GaN, SiC, BN, InSe, MoS2, BaTiO3, CsPbI3. If there are multiple values, please give me the largest value. | The maximum decomposition enthalpies for the requested materials are as follows: - Ag2O3: 0.2565836414999989 eV/atom - MgZn2: 0.1356 eV/atom - GaAs: 0.075 eV/atom - GaN: 0.07162 eV/atom - SiC: -0.172 eV/atom - BN: -0.358 eV/atom - InSe: -0.01816 eV/atom - MoS2: 0.267 eV/atom - BaTiO3: 3.055 eV/atom - CsPbI3: 0.017536019000000902 eV/atom Please note that the values for SiC, BN, and InSe are negative, indicating that the decomposition is exothermic. All other values are positive, indicating endothermic decompositions. | The decomposition enthalpies (in kJ/mol) of the following materials are:
Ag2O3: -335
MgZn2: -82
GaAs: -317
GaN: -505
SiC: -40
BN: -248
InSe: -125
MoS2: -393
BaTiO3: -112
CsPbI3: -153 |
| B6.5 | Give me the formation energies of the following materials: Ag2O3, MgZn2, GaAs, GaN, SiC, BN, InSe, MoS2, BaTiO3, CsPbI3. If there are multiple values, please give me the smallest value. | The smallest formation energies of the materials you asked for are as follows: Ag2O3: -0.3601 eV/atom, MgZn2: -0.1306 eV/atom, GaAs: -0.446 eV/atom, GaN: -0.052 eV/atom, SiC: -0.204 eV/atom, BN: -1.41 eV/atom, InSe: -0.742582154921875 eV/atom, MoS2: -1.202 eV/atom, BaTiO3: -3.493 eV/atom, CsPbI3: -1.296 eV/atom. | Ag2O3: -2.63 eV/atom MgZn2: -0.27 eV/atom GaAs: -4.13 eV/atom GaN: -3.51 eV/atom SiC: -4.98 eV/atom BN: -3.8 eV/atom InSe: -0.89 eV/atom MoS2: -3.31 eV/atom BaTiO3: -4.45 eV/atom CsPbI3: -2.32 eV/atom |

Table 6 – continued from previous page

| | Question | Answer | |
| --- | --- | --- | --- |
| | | **LLaMP** | **GPT-3.5** |
| B6.6 | Give me the decomposition enthalpies of the following materials: Ag2O3, MgZn2, GaAs, GaN, SiC, BN, InSe, MoS2, BaTiO3, CsPbI3. If there are multiple values, please give me the smallest value. | The smallest decomposition enthalpy values for the requested materials are as follows: Ag2O3: 0.019373648083333528 eV/atom, MgZn2: -0.0197 eV/atom, GaAs: -0.446 eV/atom, GaN: -0.05231 eV/atom, SiC: -0.204 eV/atom, BN: -1.41 eV/atom, InSe: -0.06336 eV/atom, MoS2: -0.669 eV/atom, BaTiO3: -0.063 eV/atom, CsPbI3: -0.018602 eV/atom. | The decomposition enthalpies of the materials are as follows: - Ag2O3: -1138.5 kJ/mol - MgZn2: -54.2 kJ/mol - GaAs: -452.2 kJ/mol - GaN: -510.9 kJ/mol - SiC: -2133.6 kJ/mol - BN: -2551.0 kJ/mol - InSe: -209.3 kJ/mol - MoS2: -739.3 kJ/mol - BaTiO3: -393.5 kJ/mol - CsPbI3: -65.3 kJ/mol |
| B6.7 | Give me the energy above hull values of the following materials: Ba2MnSe3, KAlSiO4, LiCuO, H8C3N2O, Dy2ZnS4, Rb2BeF4, K9Fe2S7, Tb2FeC4, ScAlAu, CaCrO4. If there are multiple values, please give me the lowest one. | Here are the lowest energy above hull values for the requested materials: Ba2MnSe3: 0.0 eV/atom KAlSiO4: 0.0 eV/atom LiCuO: 0.0 eV/atom H8C3N2O: 0.0735 eV/atom Dy2ZnS4: 0.00966 eV/atom Rb2BeF4: 0.0 eV/atom K9Fe2S7: 0.0 eV/atom Tb2FeC4: 0.0 eV/atom ScAlAu: 0.0 eV/atom CaCrO4: 0.0 eV/atom. Please note that these are the lowest values found in the Materials Project database and actual values may vary based on different entries. | The energy above hull values for the given materials are as follows: - Ba2MnSe3: 0.0000 eV/atom - KAlSiO4: 0.0000 eV/atom - LiCuO: 0.0000 eV/atom - H8C3N2O: 0.0000 eV/atom - Dy2ZnS4: 0.0000 eV/atom - Rb2BeF4: 0.0000 eV/atom - K9Fe2S7: 0.0000 eV/atom - Tb2FeC4: 0.0000 eV/atom - ScAlAu: 0.0000 eV/atom - CaCrO4: 0.0000 eV/atom |

Table 6 – continued from previous page

| | Question | Answer | |
| | | LLaMP | GPT-3.5 |
|---|---|---|---|
| B6.8 | What are the bandgaps of the following materials: Ba(PdS2)2, FePO4, DyBi2IO4, NaFe(SO4)2, Eu2B5BrO9, Sr5As3H, Tl-GaS2, AgSbS2, Pr3S3N, LiLaO2. If there are polymorphs, give me only the bandgap of the stable material. | The bandgaps for the materials are:
- Ba(PdS2)2: 0.779 eV
- Most stable FePO4: 2.1932 eV
- DyBi2IO4: 1.367 eV
- Most stable NaFe(SO4)2: 2.094 eV
- Eu2B5BrO9: 0.88 eV
- Sr5As3H: 0.676 eV
- TlGaS2: 1.7885 eV
- Most stable AgSbS2: 1.372 eV
- Pr3S3N: 1.15 eV
- Most stable LiLaO2: 3.7336 eV
Please note that these values are obtained from the Materials Project database and might be different in experimental results due to various factors. | I don't have real-time data access, and my training only includes information available up to January 2022. As a result, I may not have specific bandgap values for the materials you mentioned. However, I can provide you with general information about some of these materials and their known properties up to my last update:
Ba(PdS2)2: I don't have specific bandgap information for this compound. The bandgap may depend on factors such as crystal structure and synthesis conditions.
FePO4: Iron(II) phosphate, known as ferrous phosphate, typically has a bandgap in the range of 2.5-3.0 eV.
DyBi2IO4: Dysprosium bismuth iodate. I don't have specific bandgap information for this compound.
NaFe(SO4)2: Sodium iron sulfate. I don't have specific bandgap information for this compound.
Eu2B5BrO9: Europium borate bromate. I don't have specific bandgap information for this compound.
Sr5As3H: Strontium arsenide hydride. I don't have specific bandgap information for this compound.
TlGaS2: Thallium gallium sulfide. I don't have specific bandgap information for this compound.
AgSbS2: Silver antimony sulfide. This material is a semiconductor with a bandgap typically around 1.5-2.0 eV.
Pr3S3N: Praseodymium sulfide nitride. I don't have specific bandgap information for this compound.
LiLaO2: Lithium lanthanum oxide. I don't have specific bandgap information for this compound.
The most stable structure for each material can vary based on synthesis conditions and other factors. For accurate and up-to-date bandgap values and structural information, you may want to consult recent scientific literature or databases that specialize in material properties. |

Table 6 – continued from previous page

| | Question | Answer | |
| | | LLaMP | GPT-3.5 |
|---|---|---|---|
| B6.9 | How to synthesize $YMnO_3$? | To synthesize $YMnO_3$, a solid-state reaction method can be employed. Here is a summarized procedure based on the synthesis details provided in the literature with DOI: 10.1021/ic502027k:
1. Dissolve Y(NO3)3·6H2O, Mn(NO3)2·4H2O, and a source of [OH-] in water to create a homogeneous solution.
2. Add citric acid to the solution to act as a mixing media.
3. Heat the mixture at 250°C for 2 hours to achieve pre-calcination.
4. Finally, heat the pre-calcined mixture at a temperature range of 700°C to 930°C for 10 to 18 hours in an air atmosphere to form $YMnO_3$. | Synthesizing $YMnO_3$ (yttrium manganite) typically involves the solid-state reaction method. Please note that the details may vary depending on the specific conditions and equipment available in your laboratory. Here's a general procedure:
**Materials:** Yttrium oxide ($Y_2O_3$) Manganese oxide ($MnO_2$ or $Mn_2O_3$) High-purity manganese carbonate ($MnCO_3$) Lithium carbonate ($Li_2CO_3$) or lithium hydroxide (LiOH) Crucible and pestle High-temperature furnace Mortar and pestle Platinum crucible or alumina crucible
**Procedure:**
**Step 1: Weighing the Precursors**
Weigh the appropriate amounts of yttrium oxide ($Y_2O_3$), manganese oxide ($MnO_2$ or $Mn_2O_3$), and lithium carbonate (Li2CO3) or lithium hydroxide (LiOH) according to the stoichiometric ratio of YMnO3.
**Step 2: Mixing the Powders** 2. GPT-3.5 starts to fail numbering correctly in this response. Grind the powders thoroughly in a mortar and pestle to ensure a homogenous mixture.
**Step 3: Heating** 3. Place the mixed powder in a crucible or alumina crucible. Heat the mixture in a high-temperature furnace. The specific temperature and heating rate depend on the exact synthesis conditions, but a typical range is 800-1100°C.
**Step 4: Sintering** 5. Hold the mixture at the target temperature for a specific duration (several hours) to allow for sintering and the formation of $YMnO_3$.
**Step 5: Cooling** 6. Slowly cool the sample to room temperature to avoid thermal stress.
**Step 6: Characterization** 7. Characterize the synthesized $YMnO_3$ using techniques such as X-ray diffraction (XRD), scanning electron microscopy (SEM), and energy-dispersive X-ray spectroscopy (EDS) to confirm the desired phase and composition.
**Notes:** The choice of manganese precursor ($MnO_2$ or $Mn_2O_3$) and lithium source (carbonate or hydroxide) can impact the synthesis process and may require optimization. It's crucial to use high-purity starting materials to avoid impurities in the final product. The specific temperatures and times mentioned are general guidelines and may need to be adjusted based on the equipment and conditions in your laboratory. Always follow safety protocols when working with high temperatures and potentially hazardous materials. This is a simplified overview, and you should consult relevant literature and adapt the procedure based on your specific requirements and equipment. |

Table 6 – continued from previous page

| | Question | Answer | |
| | | LLaMP | GPT-3.5 |
|---|---|---|---|
| B6.10 | How to synthesize LiFePO$_4$? Please provide a detailed step-by-step procedure and the reference. | LiFePO$_4$ can be synthesized using various methods. One method involves a high-temperature solid-state synthesis using precursors such as Fe$_2$O$_3$, LiOH, and NH$_4$H$_2$PO$_4$. The precursors are mixed and heated at temperatures up to 850°C for several days. Another method uses a sol-gel combustion approach with precursors like FePO$_4$ and Li$_2$CO$_3$, where the mixture is calcined at 700°C in a nitrogen atmosphere. For detailed procedures, please refer to the following scientific literature: 1. High-temperature solid-state synthesis - DOI: 10.1021/jp407510h 2. Ion-exchange method - DOI: 10.1021/cm402617b 3. Solution method - DOI: 10.1021/jp306936t 4. Solid-state reactions - DOI: 10.1021/cm303259j 5. Sol-gel combustion method - DOI: 10.1007/s10853-011-6139-7 | Synthesizing LiFePO$_4$ (lithium iron phosphate) typically involves a solid-state reaction method. Below is a step-by-step procedure for synthesizing LiFePO$_4$: Materials: - Lithium carbonate (Li2CO3) - Iron (II) acetate (Fe(CH$_3$COO)$_2$) - Ammonium dihydrogen phosphate (NH4H2PO4) - Phosphoric acid (H3PO4) - Deionized water - Ethanol - Mortar and pestle - Crucible - Furnace Procedure: 1. Weigh out the appropriate amounts of lithium carbonate (Li2CO3) and iron (II) acetate (Fe(CH$_3$COO)$_2$) in a 1:1 molar ratio. For example, if you want to synthesize 1 mole of LiFePO$_4$, you would weigh out 74.55 grams of Li$_2$CO$_3$ and 173.93 grams of Fe(CH$_3$COO)$_2$. 2. Grind the lithium carbonate and iron (II) acetate together using a mortar and pestle to ensure a homogenous mixture. 3. Weigh out the appropriate amount of ammonium dihydrogen phosphate (NH4H2PO4) and add it to the lithium carbonate and iron (II) acetate mixture in a 1:1 molar ratio. For example, if you are synthesizing 1 mole of LiFePO$_4$, you would weigh out 122.00 grams of NH4H2PO4. 4. Add a few drops of phosphoric acid (H3PO4) to the mixture to adjust the pH to around 7. This will help in the formation of LiFePO$_4$. 5. Add a small amount of deionized water to the mixture and mix it thoroughly to form a slurry. 6. Slowly evaporate the water from the slurry by placing it on a hot plate or in a furnace at a low temperature. 7. Once the water has evaporated, heat the mixture in a crucible in a furnace at around 600-700°C for several hours to initiate the solid-state reaction and form LiFePO$_4$. 8. After the reaction is complete, allow the crucible to cool and collect the LiFePO$_4$ powder. 9. Wash the LiFePO$_4$ powder with ethanol to remove any impurities and then dry it in a desiccator. Reference: P. G. Bruce, B. Scrosati, J.-M. Tarascon, "Nanomaterials for Rechargeable Lithium Batteries," Angew. Chem. Int. Ed. 2008, 47, 2930-2946. DOI: 10.1002/anie.200702505. |

Table 6 – continued from previous page

| | Question | Answer | |
|---|---|---|---|
| | | **LLaMP** | **GPT-3.5** |
| B6.11 | Could you insert one Li atom at the tetrahedral interstitial site in diamond cubic Si? Pull out the most stable DC Si structure from MP directly in JSON string, modify the retrieved structure directly, and give me the final pymatgen structure as a JSON string. | The final structure after inserting a Li atom at the tetrahedral interstitial site in the diamond cubic Si structure is as follows: | Here's the JSON string for the most stable diamond cubic Si structure retrieved directly from MP: |

LLaMP:

```
{'@module': 'pymatgen.core.structure', '
    @class': 'Structure', 'charge': 0,
    'lattice': {'matrix': [[3.333573,
    0.0, 1.924639], [1.111191,
    3.142924, 1.924639], [0.0, 0.0,
    3.849278]], 'pbc': [True, True,
    True], 'a': 3.8492784033699095, 'b
    ': 3.8492794116013456, 'c':
    3.849278, 'alpha':
    60.00001213094421, 'beta':
    60.00000346645984, 'gamma':
    60.00001097545789, 'volume':
    40.32952684741405}, 'properties':
    {}, 'sites': [{'species': [{'
    element': 'Si', 'occu': 1}], 'abc':
    [0.875, 0.875, 0.875], 'xyz':
    [3.8891685, 2.7500584999999997,
    6.7362365], 'properties': {'magmom
    ': -0.0}, 'label': 'Si'}, {'species
    ': [{'element': 'Si', 'occu': 1}],
    'abc': [0.125, 0.125, 0.125], 'xyz
    ': [0.5555955, 0.3928655,
    0.9623195], 'properties': {'magmom
    ': -0.0}, 'label': 'Si'}, {'species
    ': [{'element': 'Li', 'occu': 1}],
    'abc': [0.5, 0.5, 0.5], 'xyz':
    [2.222382, 1.571462, 1.924639], '
    properties': {}, 'label': 'Li'}]}
```

GPT-3.5:

```json
{'sites': [{'species': [{'element': 'Si', 'occu': 1}],
  'abc': [0, 0, 0], 'xyz': [0, 0, 0], 'label': 'Si', 'properties': {}},
  {'species': [{'element': 'Si', 'occu': 1}],
  'abc': [0.25, 0.25, 0.25], 'xyz': [0.178, 0.178, 0.178], 'label': 'Si', 'properties':
    {}}],
  'lattice': {'matrix': [[3.866002, 0, 2.243049],
  [3.866002, 3.866002, 0],
  [0, 3.866002, 2.243049]],
  'a': 5.431, 'b': 5.431, 'c': 5.431,
  'alpha': 109.471219, 'beta': 109.471219, 'gamma': 109.471219,
  'volume': 129.584354898},
  'properties': {'space_group': {'number': 227,
  'source': 'spglib',
  'symbol': 'Fd-3m'}}}
```

Now, let's insert one Li atom at the tetrahedral interstitial site in this diamond cubic Si structure and retrieve the final pymatgen structure as a JSON string:

```json
{'sites': [{'species': [{'element': 'Si', 'occu': 1}],
  'abc': [0, 0, 0],
  'xyz': [0, 0, 0],
  'label': 'Si',
  'properties': {}},
  {'species': [{'element': 'Si', 'occu': 1}],
  'abc': [0.25, 0.25, 0.25],
  'xyz': [0.178, 0.178, 0.178],
  'label': 'Si',
  'properties': {}},
  {'species': [{'element': 'Li', 'occu': 1}],
  'abc': [0.5, 0.5, 0.5],
  'xyz': [0.5, 0.5, 0.5],
  'label': 'Li',
  'properties': {}}],
  'lattice': {'matrix': [[3.866002, 0, 2.243049],
  [3.866002, 3.866002, 0],
  [0, 3.866002, 2.243049]],
  'a': 5.431,
  'b': 5.431,
  'c': 5.431,
  'alpha': 109.471219,
  'beta': 109.471219,
  'gamma': 109.471219,
  'volume': 129.584354898},
  'properties': {'space_group': {'number': 227,
  'source': 'spglib',
  'symbol': 'Fd-3m'}}}
```

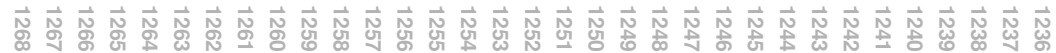

Table 6 – continued from previous page

| | Question | Answer | |
| --- | --- | --- | --- |
| | | **LLaMP** | **GPT-3.5** |
| B6.12 | What is the magnetic ordering of stable $FePO_4$? Give me its material id, space group, and magnetization per formula unit. | The stable $FePO_4$ has a ferromagnetic (FM) magnetic ordering. Its material ID is 'mp-753395', it belongs to the Triclinic crystal system with space group symbol 'P-1' and number 2. The total magnetization per formula unit is 5.00000015 Bohr magnetons. | The magnetic ordering of stable $FePO_4$ is antiferromagnetic. Its material id is mp-5405, and its space group is Pnma. The magnetization per formula unit is 4.14 μB. |

## C   OTHER CAPABILITIES

### C.1   AUTOMATED COMPUTATIONAL WORKFLOW

The following example shows how LLaMP uses computational workflow `atomate2` and job manager `jobflow` to run automated molecular dynamics simulation using foundation machine learning interatomic potential MACE-MP-0 (Batatia et al., 2023).

```
Query:
I would like you to help me run MD simulation with the follwing steps:

1. Retrieve the stable LiTaO3 structure from Materials Project
2. Create a supercell around the size of 100 atoms
3. Use atomate2 with foundation ML forcefield MACE to run Nose-Hoover NVT simulation
   for 2 ps with annealing temperature scheduling 300 to 800 back to 300 Kelvin.
   Use 2 fs timestep and save the trajectory to a ASE traj file.
4. Read the trajectory file and plot the system temperature vs time

> Entering new AgentExecutor chain...
Thought: Do I need to use a tool? Yes
Action: MPStructureRetriever
Action Input: LiTaO3

> Entering new AgentExecutor chain...
Action:
```json
{
  "action": "search_materials_structure__get",
  "action_input": {
    "formula": "LiTaO3",
    "limit": 5,
    "fields": "material_id,structure"
  }
}
```

```
All retrieved structures are saved as Pymatgen Structure JSON files to the following
    paths: mp-3666.json, mp-1105280.json, mp-754345.json, mp-1105216.json, mp
    -1105326.json

> Finished chain.
{'input': 'LiTaO3', 'output': 'All retrieved structures are saved as Pymatgen
    Structure JSON files to the following paths: mp-3666.json, mp-1105280.json, mp
    -754345.json, mp-1105216.json, mp-1105326.json'}
Do I need to use a tool? Yes
Action: MPSummaryExpert
Action Input: mp-3666

> Entering new AgentExecutor chain...
Action:
```json
{
  "action": "search_materials_summary__get",
  "action_input": {
    "material_ids": "mp-3666"
  }
}
```Error on search_materials_summary__get: 'fields' must be specified in the query.
    Please revise arguments or try smaller request by specifying 'limit' in request
    .
```json
{
```

```
      "action": "search_materials_summary__get",
      "action_input": {
        "material_ids": "mp-3666",
        "fields": "material_id,formula_pretty,composition,nsites,symmetry"
      }
    }
```

```
[{'nsites': 10, 'composition': {'Li': 2.0, 'Ta': 2.0, 'O': 6.0}, 'formula_pretty': '
    LiTaO3', 'symmetry': {'crystal_system': 'Trigonal', 'symbol': 'R3c', 'number':
    161, 'point_group': '3m', 'symprec': 0.1, 'version': '2.0.2'}, 'material_id': '
    mp-3666'}]
Action:
```json
{
  "action": "Final Answer",
  "action_input": "Material ID mp-3666 corresponds to Lithium Tantalate (LiTaO3). It
      has a trigonal crystal system with space group R3c (number 161) and point
      group 3m. The structure consists of 10 sites, composed of Li, Ta, and O."
}
```
Finished chain.
{'input': 'mp-3666', 'output': 'Material ID mp-3666 corresponds to Lithium Tantalate
    (LiTaO3). It has a trigonal crystal system with space group R3c (number 161)
    and point group 3m. The structure consists of 10 sites, composed of Li, Ta, and
    O.'}
Do I need to use a tool? Yes
Action: MPStructureRetriever
Action Input: mp-3666

Entering new AgentExecutor chain...
Action:
```json
{
  "action": "search_materials_structure__get",
  "action_input": {
    "material_ids": "mp-3666",
    "limit": 1,
    "fields": "material_id,structure"
  }
}
```
```

```
All retrieved structures are saved as Pymatgen Structure json files to the following
    paths: mp-3666.json

Do I need to use a tool? Yes
Action: Python_REPL
Action Input:
```

```python
from pymatgen.core import Structure

# Load the structure from the JSON file
structure = Structure.from_file('mp-3666.json')

# Create a supercell to approximately reach 100 atoms
num_atoms = len(structure)
multiplier = int((100 / num_atoms) ** (1/3)) + 1 # Calculate the multiplier for each
    dimension
```

```python
supercell = structure * (multiplier, multiplier, multiplier)

# Save the supercell structure to a file and print the number of atoms
supercell.to(fmt="poscar", filename="supercell.vasp")
print({"supercell_path": "supercell.vasp", "num_atoms": len(supercell)})
```

```
{'supercell_path': 'supercell.vasp', 'num_atoms': 270}
```

```
Do I need to use a tool? Yes
Action: MLFFMDAgent
Action Input: {"structure_path": "supercell.vasp", "forcefield": "MACE", "
    simulation_type": "NVT", "temperature_schedule": [300, 800, 300], "
    simulation_time": 2, "timestep": 2, "trajectory_file": "trajectory.traj"}Do I
    need to use a tool? Yes

> Entering new AgentExecutor chain...
Action:
```json
{
  "action": "MLFF MD",
  "action_input": {
    "run_mode": "local",
    "project": "llamp-atomate2",
    "atom_path_or_dict": "supercell.vasp",
    "force_field_name": "MLFF.MACE",
    "time_step": 2,
    "n_steps": 1000,
    "ensemble": "NVT",
    "dynamics": "langevin",
    "temperature": [300, 800, 300],
    "traj_file": "trajectory.traj",
    "traj_file_fmt": "ase",
    "traj_interval": 1
  }
}
```Action:
```

```
2024-05-12 14:42:31,797 INFO Started executing jobs locally
2024-05-12 14:42:31,837 INFO Starting job - Forcefield MD (040866d1-fbcb-4e0b-9044-0
    ed3cf9d0eef)
Using Materials Project MACE for \texttt{MACECalculator} with /global/homes/c/
    cyrusyc/.cache/mace/5yyxdm76
Using \texttt{float32} for \texttt{MACECalculator}, which is faster but less
    accurate. Recommended for MD. Use float64 for geometry optimization.
Default dtype float32 does not match model dtype float64, converting models to
    float32.
2024-05-12 14:43:58,683 INFO Finished job - Forcefield MD (040866d1-fbcb-4e0b-9044-0
    ed3cf9d0eef)
INFO:jobflow.core.job:Finished job - Forcefield MD (040866d1-fbcb-4e0b-9044-0
    ed3cf9d0eef)
2024-05-12 14:43:58,710 INFO Finished executing jobs locally
INFO:jobflow.managers.local:Finished executing jobs locally
{'energy': -2262.60595703125, 'n_steps': 1001}
```

```
Action:
```json
{
  "action": "Final Answer",
  "action_input": "The molecular dynamics simulation was successfully completed. The
      final energy of the system after 1001 steps is -2262.60595703125 eV."
}
```Action:
```json
{
  "action": "Final Answer",
  "action_input": "The molecular dynamics simulation was successfully completed. The
      final energy of the system after 1001 steps is -2262.60595703125 eV."
}
```

> Finished chain.
{'input': '{"structure_path": "supercell.vasp", "forcefield": "MACE", "
    simulation_type": "NVT", "temperature_schedule": [300, 800, 300], "
    simulation_time": 2, "timestep": 2, "trajectory_file": "trajectory.traj"}', '
    output': 'The molecular dynamics simulation was successfully completed. The
    final energy of the system after 1001 steps is -2262.60595703125 eV.'}Do I need
     to use a tool? Yes
Action: Python_REPL
Action Input:
```

```python
import matplotlib.pyplot as plt
from ase.io import read

# Load the trajectory
trajectory = read('trajectory.traj', index=':')

# Extract temperatures and time steps
temperatures = [step.get_temperature() for step in trajectory]
time_steps = [i * 2 for i in range(len(temperatures))] # Calculate time steps
    manually since each step is 2 fs

# Plotting
plt.figure(figsize=(10, 5))
plt.plot(time_steps, temperatures, label='Temperature (K)', marker='o')
plt.title('Temperature vs Time during MD Simulation')
plt.xlabel('Time (fs)')
plt.ylabel('Temperature (K)')
plt.grid(True)
plt.legend()
plt.savefig('temperature_vs_time.png')
plt.show()
print({"temperature_vs_time_plot": "temperature_vs_time.png"})
```

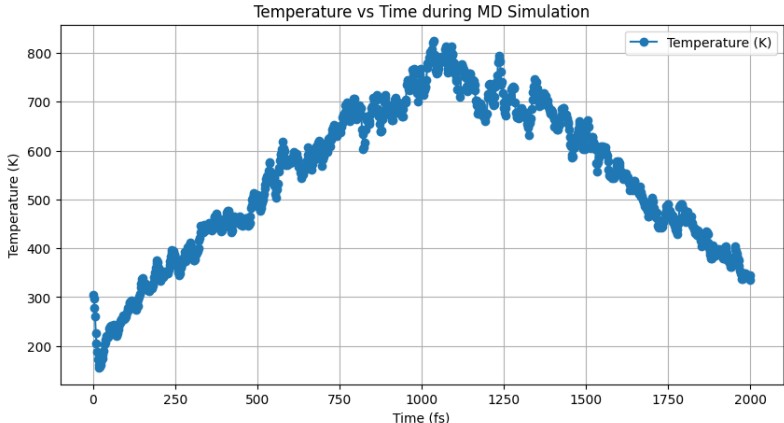

## C.2 SIMULATION CODE GENERATION AND EXECUTION

Below we present how LLaMP can generate, self-correct, and execute ASE simulation code to calculate the bulk modulus of aluminum.

```
Generate a script and run ASE EMT calculator to relax 2x2x2 Al supercell and
    calculate its bulk modulus

# Entering new AgentExecutor chain...
To generate a script and run the ASE (Atomic Simulation Environment) EMT (Effective
    Medium Theory) calculator to relax a 2x2x2 Al supercell and calculate its bulk
    modulus, we need to write a Python script that uses ASE's functionalities. Here
    's a step-by-step breakdown of what the script will do:

1. Import necessary modules from ASE.
2. Create an aluminum (Al) supercell.
3. Set up the EMT calculator.
4. Relax the structure using the EMT calculator.
5. Calculate the bulk modulus.

Let's write the script:
```

```python
from ase import Atoms
from ase.build import bulk
from ase.calculators.emt import EMT
from ase.constraints import StrainFilter, ExpCellFilter
from ase.optimize import BFGS
from ase.eos import EquationOfState

# Create an Al supercell
al = bulk('Al', 'fcc', a=4.05, cubic=True)
al = al.repeat((2, 2, 2)) # Repeat the unit cell to create a 2x2x2 supercell

# Set up the EMT calculator
al.set_calculator(EMT())

# Relax the structure
strain_filter = StrainFilter(al)
optimizer = BFGS(strain_filter)
optimizer.run(fmax=0.01)

# Calculate the bulk modulus
```

```
volumes = []
energies = []
for x in [0.98, 0.99, 1.00, 1.01, 1.02]:
    al_copy = al.copy()
    al_copy.set_calculator(EMT())
    cell = al_copy.get_cell()
    al_copy.set_cell(cell * x, scale_atoms=True)
    volumes.append(al_copy.get_volume())
    energies.append(al_copy.get_potential_energy())

eos = EquationOfState(volumes, energies)
v0, e0, B = eos.fit()
eos.plot('al-eos.png')

print(f'Bulk_modulus:_{B_/_kJ}_GPa')
```

Now, let's execute this script using the Python\_REPL tool.

Action: Python\_REPL
Action Input:

```
from ase import Atoms
from ase.build import bulk
from ase.calculators.emt import EMT
from ase.constraints import StrainFilter, ExpCellFilter
from ase.optimize import BFGS
from ase.eos import EquationOfState
from ase.units import kJ

# Create an Al supercell
al = bulk('Al', 'fcc', a=4.05, cubic=True)
al = al.repeat((2, 2, 2)) # Repeat the unit cell to create a 2x2x2 supercell

# Set up the EMT calculator
al.set_calculator(EMT())

# Relax the structure
strain_filter = StrainFilter(al)
optimizer = BFGS(strain_filter)
optimizer.run(fmax=0.01)

# Calculate the bulk modulus
volumes = []
energies = []
for x in [0.98, 0.99, 1.00, 1.01, 1.02]:
    al_copy = al.copy()
    al_copy.set_calculator(EMT())
    cell = al_copy.get_cell()
    al_copy.set_cell(cell * x, scale_atoms=True)
    volumes.append(al_copy.get_volume())
    energies.append(al_copy.get_potential_energy())

eos = EquationOfState(volumes, energies)
v0, e0, B = eos.fit()

print(f'Bulk_modulus:_{B_/_kJ}_GPa')
```

```
        Step     Time          Energy         fmax
BFGS:    0 01:37:31     -0.048066      8.967521
BFGS:    1 01:37:31      1.542253     33.456077
BFGS:    2 01:37:31     -0.154186      1.250352
BFGS:    3 01:37:31     -0.156157      0.259261
BFGS:    4 01:37:31     -0.156246      0.000522
Bulk modulus: 3.923492594026645e-23 GPa
```

The output of the bulk modulus calculation seems to be incorrect because the value
    is extremely small and not in the expected range for aluminum. This is likely
    due to a unit conversion error. The ASE package uses eV and Angstroms for
    energy and length, respectively, and the bulk modulus should be in units of eV/
    Angstrom^3, which can be converted to GPa. The conversion factor from eV/
    Angstrom^3 to GPa is approximately 160.21766208. Let's correct the script to
    include the proper conversion factor and rerun the calculation.

Action: Python\_REPL
Action Input:

```python
from ase import Atoms
from ase.build import bulk
from ase.calculators.emt import EMT
from ase.constraints import StrainFilter, ExpCellFilter
from ase.optimize import BFGS
from ase.eos import EquationOfState
from ase.units import kJ

# Create an Al supercell
al = bulk('Al', 'fcc', a=4.05, cubic=True)
al = al.repeat((2, 2, 2)) # Repeat the unit cell to create a 2x2x2 supercell

# Set up the EMT calculator
al.set_calculator(EMT())

# Relax the structure
strain_filter = StrainFilter(al)
optimizer = BFGS(strain_filter)
optimizer.run(fmax=0.01)

# Calculate the bulk modulus
volumes = []
energies = []
for x in [0.98, 0.99, 1.00, 1.01, 1.02]:
    al_copy = al.copy()
    al_copy.set_calculator(EMT())
    cell = al_copy.get_cell()
    al_copy.set_cell(cell * x, scale_atoms=True)
    volumes.append(al_copy.get_volume())
    energies.append(al_copy.get_potential_energy())

eos = EquationOfState(volumes, energies)
v0, e0, B = eos.fit()

# Convert the bulk modulus from eV/Angstrom^3 to GPa
B_GPa = B * 160.21766208

print(f'Bulk modulus: {B_GPa} GPa')
```

```
       Step     Time          Energy          fmax
BFGS:    0 01:37:48      -0.048066       8.967521
BFGS:    1 01:37:48       1.542253      33.456077
BFGS:    2 01:37:48      -0.154186       1.250352
BFGS:    3 01:37:48      -0.156157       0.259261
BFGS:    4 01:37:48      -0.156246       0.000522
Bulk modulus: 39.23492594026644 GPa
```

```
Do I need to use a tool? No
Final Answer: The corrected bulk modulus of the relaxed 2x2x2 Al supercell
    calculated using the ASE EMT calculator is approximately 39.23 GPa.
# Finished chain.
```

