# OpenReview forum: "LLaMP: Large Language Model Made Powerful for High-fidelity Materials Knowledge Retrieval"
_ICLR.cc/2025/Conference — Submitted to ICLR 2025_

### Official Review · Reviewer_PUgh · 2024-11-01

**Soundness:** 3
**Presentation:** 3
**Contribution:** 3
**Rating:** 8
**Confidence:** 4

**Summary:**

The paper proposes LLaMP, a RAG framework based on hierarchical ReAct agents. The supervisor agent handles high level logic while assistant agents interact with Materials Project, arXiv, Wikipedia and atomistic simulation tools.

The authors then introduce a metric, the Self-Consistency of Response (SCoR), which aggregates the model's confidence in answering with the variability of the answers.

Experiments show that LLaMP achieves a low MAE and high SCoR in generating Bulk Modulus, Formation Energy, and Electronic Bandgap while LLMs and other RAG frameworks achieve higher MAEs and comparable or lower SCoR.

LLaMP is also able to predict the magnetic orderings and total magnetisation, retrieve inorganic synthesis recipes, edit crystal structures and drive simulations.

**Strengths:**

**Originality**
LLaMP is original in its use of hierarchical agents and the combination of agents used. The metrics and combination of tasks is also original.

**Quality**
The hierarchical orchestration is well thought out and seems to perform well. The variety of agents makes the framework useful for materials scientists. The SCoR is mostly well justified. The experiments are convincing. The variety of tasks used is a bonus.

**Clarity**
The paper is easy to read  and well organised. Figures and Tables are informative (except Figure 2, see Weaknesses).

**Significance**
Accuracy of answers and reducing hallucinations is important when using LLMs in a scientific context. LLaMP is a significant step towards addressing these issues and thus is a valuable contribution to the community.

**Weaknesses:**

1) The name precision in SCoR feels misleading. When SCoR = 1, the model generates the same response to queries, meaning that the standard deviation is zero and also the precision is zero. This is counter-intuitive as I think of precision as True Positives / (True Positives + False Positives).
2) Figure 2 is unclear. How should I read the values produced by each method? What is the ground truth?
3) (Minor) I can't find a citation for LangChain.

**Questions:**

1) Can you please elaborate on how the standard deviation in SCoR is calculated? I assume that it only works for numerical answers and it's the standard deviation of the population.
2) Can you please clarify Figure 2. What are the ground truth values?
3) Can you provide code to reproduce the framework? This would add value to your submission.

---

> ### Author Response · Authors · 2024-11-15
>
> Thank you for reviewing our paper and recognizing our work. We answer your questions and comments below.
>
> **W1:**
>
> We apologize for the confusion. As we are quantifying the numerical prediction performance, the “Precision” used across this work is “sampled standard deviation” in the field of measurement/engineering (https://en.wikipedia.org/wiki/Accuracy_and_precision) and is termed “variability” or “random error” in statistics. We appreciate the reviewer’s suggestion and have revised the text and caption where appropriate to minimize the confusion.
>
> **W2, Q2:**
>
> We have updated Figure 2 to better visualize the boxplot and added the ground truth values as horizontal lines. The short summary of each plot and table is added as well. Thank you for the suggestion!
>
> **W3:**
>
> We have added the citation of LangChian back. Thank you!
>
> **Q1:**
>
> Yes, it’s the standard deviation of population and it measure the numerical response.
>
>
> **Q3:**
>
> We will release the code along with example notebooks to reproduce the result on github. We also upload the code to the supplementary materials for your reference.

---

> > ### Comment · Reviewer_PUgh · 2024-11-27
> >
> > Thank you for responding to the comments and releasing the code. Clarifying precision as sample standard deviation helps. Figure 2 is now clearer.

---

### Official Review · Reviewer_X8Ga · 2024-11-03

**Soundness:** 1
**Presentation:** 2
**Contribution:** 2
**Rating:** 1
**Confidence:** 4

**Summary:**

The paper proposes a new method for LLMs (LLAMP) applied to materials science based on retrieval-augmented generation (RAG) and hierarchical reasoning-and-acting (ReAct) that addresses language-based materials science tasks. The papers starts by motivating LLAMP based on the fact that many LLMs lack access to up-to-data for solving timely materials challenges and are prone to hallucination. As such, LLAMP aims to alleviate those challenges by providing a way to infuse external data sources when solving materials science tasks. The paper claims five contributions: 1. the LLAMP method; 2. metrics for understanding self-consistency for LLMs; 3. a study for materials property prediction; 4. showcasing LLAMP capabilities in synthesis and materials structure generation; 5. demonstrating high-throughput atomistic simulations using LLAMP.

Next, the paper describes related work in Section 2 and Section 3 respectively focused on the materials project, NLP in science as well as the application of prompting and tool use to solve materials science challenges. Section 4 the describes the main pieces of the LLAMP-ReAct framework, including the supervisor agent and assistant agents that help perform relevant tasks. Section 4 also outlines the metrics used in subsequent analysis, including Prediction, Coefficient of Precision, Confidence and the Self-consistency of Response (SCoR). The primary aim of the metric is the understand the consistency and confidence of LLM responses to certain queries where reliability is important. Section 5 provides the main experiments with the main focus on including materials property prediction. The paper then provides an analysis based on the metrics proposed in Section 4. The analysis generally shows that LLAMP provides lower MAE in property prediction and usually second best in SCoR. Section 5.2 also showcases a study on data retrieval for materials property analysis where LLAMP with its ReAct framework generally outperforms vanilla LLMs relying on implicit knowledge. Section 5.3 provides a brief description summarization of synthesis procedures as well as crystal generation and language-drive simulation. The discussion of the experiments in Section 5.3 is brief with much reference to the appendix for supporting information. The general claims in Section 5.3 is that LLAMP outperforms vanilla LLMs.

The paper concludes in Section 6 with a discussion on robustness, which ties into one of the main motivations outlined in the introduction, as well as limitations and a broader summary of the work and potential future work.

**Strengths:**

* The paper introduces a new framework to relevant challenges in materials science.
* The tools and knowledge bases included in the LLAMP framework could be useful to build upon.

**Weaknesses:**

In its current form, the paper has substantial weaknesses related to limited machine learning novelty, scant evidence of the initial claims and missing discussion of relevant related work.

* Machine Learning Novelty: the paper does not provide good evidence for its machine learning novelty. The background and related work provided limited discussion of RAG and tool-calling in general and miss relevant work in related domains. As such it is difficult to understand the novelty of the LLAMP.
* Evidence for claims: In the current form, the main addition novelty related to LLAMP focuses on RAG and ReAct and many of the presented experiments show that a RAG-based performs better a RAG-based tasks (e.g., information retrieval) compared to a non-RAG model. As such, the only claim that is somewhat properly supported is that a RAG-based model is somewhat better and more consistent on tasks where RAG is useful, which is not particularly surprising given prior success of RAG. Additionally the paper provides scant evidence for claims 4 & 5 in the main text with much reference for the appendix. This is not (in my opinion) best practice and weakens the initial claims made in the introduction.
     * The claims related to the ReAct framework could also be strengthened by adding an ablation on tool calling - especially since the paper mentions flat tool calling.
* Related work: The paper provides a limited discussion of relevant work related to machine learning for relevant parts of the proposed method, including RAG and tool-calling. This extends to LLMs for chemistry and materials science where methods, including agent methods, have been proposed [1] [2]. The section on NLP for science could also benefit from a broader inclusion of work on applying LLMs for question-answering tasks in materials science [3] [4] [5] along with additional work on LLMs for materials science data extraction [6] [7 - mara].

[1] Ghafarollahi, Alireza, and Markus J. Buehler. "AtomAgents: Alloy design and discovery through physics-aware multi-modal multi-agent artificial intelligence." arXiv preprint arXiv:2407.10022 (2024).

[2] Zhang, Huan, et al. "Honeycomb: A flexible llm-based agent system for materials science." arXiv preprint arXiv:2409.00135 (2024).

[3] Yu Song, Santiago Miret, and Bang Liu. 2023. MatSci-NLP: Evaluating Scientific Language Models on Materials Science Language Tasks Using Text-to-Schema Modeling. In Proceedings of the 61st Annual Meeting of the Association for Computational Linguistics (Volume 1: Long Papers), pages 3621–3639, Toronto, Canada. Association for Computational Linguistics.

[4] Zaki, Mohd, and NM Anoop Krishnan. "MaScQA: investigating materials science knowledge of large language models." Digital Discovery 3.2 (2024): 313-327.

[5] Yu Song, Santiago Miret, Huan Zhang, and Bang Liu. 2023. HoneyBee: Progressive Instruction Finetuning of Large Language Models for Materials Science. In Findings of the Association for Computational Linguistics: EMNLP 2023, pages 5724–5739, Singapore. Association for Computational Linguistics.

[6] Hira, Kausik, et al. "Reconstructing the materials tetrahedron: challenges in materials information extraction." Digital Discovery 3.5 (2024): 1021-1037.

[7] Schilling-Wilhelmi, Mara, et al. "From text to insight: Large language models for materials science data extraction." arXiv preprint arXiv:2407.16867 (2024).

**Questions:**

* How are the agents distinguished from each other? Do they receive different prompts, different context? It seems like not all of them are LLM based - it would be good to describe more details on what is considered an agent and what is considered a tool.
* Have you compared to other RAG methods? It seems a bit unfair to have to benchmark LLMs solely on implicit knowledge. It would also be good to get additional context on how your proposed tool calling method compares to other methods [8]
* How limited are the proposed methods to Materials Project? What would be needed to apply it to other materials science knowledge sources?
* Could you add the main conclusions in the captions of figures and tables for greater clarity?

[8] Qin, Yujia, et al. "ToolLLM: Facilitating Large Language Models to Master 16000+ Real-world APIs." The Twelfth International Conference on Learning Representations.

---

> ### Author Response · Authors · 2024-11-15
>
> Thank you for your detailed feedback.  We address your comments below and appreciate the chance for us to detail the context on our design choices and restate the contributions of our approach.
>
> **W1:**
>
> We agree with the reviewer that RAG and ReAct have been explored in prior works. Our major contribution, however, is a practical framework for real-world scientific research applications. Our  methodological advance is the implementation of large-scale API orchestration and hierarchical planning itself and the augmentation of such a framework to one of the most coherent and comprehensive materials databases. LLaMP currently employs 10 MP API endpoints, with many of them having more than 10 arguments (https://api.materialsproject.org/docs) and complex logic that is highly challenging for a non-expert and infeasible for flat planning. We stress that implementing such a large and reliable tool is far beyond trivial and important for reproducibility and pointing the field toward the right path, which many previous and concurrent work do not fulfill and fall apart when people adopt them in the workflow.
>
> Such a framework enables LLaMP to perform complex, real-world tasks reliably like synthesis recipe recommendation, on-the-fly crystal structure editing, and access to DFT-calculated properties at the better quality—capabilities beyond purpose-trained LLMs. In Section 5.3, we show LLAMP’s foundational strengths in these areas, outperforming prompt-based method StructChem and fine-tuned model Darwin, which are designed for general use or their own specialized benchmark. Our framework highlights a potential for full-stack approach from data exploration to autonomous in-silico experiments for materials R&D.
>
> **W2:**
>
> The major aim of this work is to demonstrate the grounding performance of LLaMP on factual information and how such high-fidelity data guarantee the accuracy of downstream material science applications. We appreciate the reviewer for recognizing its real-world application capabilities. Due to the difficulty in comparing and quantifying the qualities of generated synthesis recipes and simulations, we mainly compare these results qualitatively with vanilla LLMs where appropriate and move the prolonged generation results to the appendix. We will release the pypi package and example notebooks to github as well as supplementary material for people to try out and reproduce these experiments.
>
> We conducted an ablation study comparing the full LLaMP framework using MP tools with a ReAct version using only the SerpAPI. Additionally, we tested the framework across different model backbones, demonstrating its compatibility with various model providers (LLaMA, Gemini, Sonnet), as shown in Ablation of section 5.2 and Table 4 in the appendix.
>
> We have added more explanation in Section 4.1. Flat planning is not feasible and scalable for orchestrating all MP API tools due to the context window limit. In fact our preliminary investigation indicates that flat planning breaks down and runs very slow for even a simple query. The number of tools one single agent can effectively use is also not commensurate with the diversity of our tasks.
>
> **W3:**
>
> We thank the reviewers for pointing out these works. On top of ChemCrow which pioneered an agentic framework for chemical science, we add [1] [2] in our related work section. We also include [3, 4, 5, 6, 7] in our NLP for science section to provide a broader overview of existing methods .

---

> ### Author Response · Authors · 2024-11-15
> **Continued above**
>
> **Q1:**
>
> The supervisor agent has access to the names and short descriptions of all assistant agents. Each assistant agent operates independently with specialized prompt messages that provide detailed API and tool schemas relevant to its role. To maintain separation between tasks, assistant agents do not have visibility into each other’s operations. Currently, only the MP API and simulation agents function as ReAct agents. We have included a comprehensive list of these roles in Appendix A1.
>
> **Q2:**
> We have compared our framework with other RAG methods in Ablation of section 5.2, which contains GPT+SerpAPI and one of our ablation cases. We also demonstrate that such a framework is robust across different backbone LLMs in Table 4.
>
>
> The approach in [8] uses a DFS-based Decision Tree to refine reasoning paths, helping the agent select the most effective sequence of API calls for tasks requiring diverse interactions. In contrast, our method emphasizes stability with challenging APIs by structuring agents hierarchically, assigning each API to a specific agent. This allows a high-level agent to maintain overall objectives without API schemas consuming memory resources, ensuring reliable performance in complex tasks.
>
> **Q3:**
>
> Our codebase is highly modular and scalable and has been packaged. We have wrapped 10  MP API endpoints into ReAct assistant agents. To extend this to other materials databases such as AFLOW [1], OQMD [2], NOMAD[3], the user will need to implement the respective RESTful APIs. To augment LLaMP with knowledge graphs [4,5], the user could take advantage of LangChain graph RAG capabilities. We have added these citations to our paper.
>
> [1] Curtarolo, S., Setyawan, W., Hart, G. L., Jahnatek, M., Chepulskii, R. V., Taylor, R. H., ... & Morgan, D. (2012). AFLOW: An automatic framework for high-throughput materials discovery. Computational Materials Science, 58, 218-226.
> [2] Kirklin, S., Saal, J. E., Meredig, B., Thompson, A., Doak, J. W., Aykol, M., ... & Wolverton, C. (2015). The Open Quantum Materials Database (OQMD): assessing the accuracy of DFT formation energies. npj Computational Materials, 1(1), 1-15.
> [3] Scheidgen, M., Himanen, L., Ladines, A. N., Sikter, D., Nakhaee, M., Fekete, Á., ... & Draxl, C. (2023). NOMAD: A distributed web-based platform for managing materials science research data. Journal of Open Source Software, 8(90), 5388.
> [4] Venugopal, V., & Olivetti, E. (2024). MatKG: An autonomously generated knowledge graph in Material Science. Scientific Data, 11(1), 217.
> [5] Ye, Y., Ren, J., Wang, S., Wan, Y., Razzak, I., Xie, T., & Zhang, W. (2024). Construction of Functional Materials Knowledge Graph in Multidisciplinary Materials Science via Large Language Model. arXiv preprint arXiv:2404.03080.
>
>
> **Q4:**
>
> We have added a short conclusion and description for all the figures and tables to enhance the readability. Thank you for the suggestion.

---

> ### Comment · Reviewer_X8Ga · 2024-11-17
>
> I appreciate the author's clarification of various details and making edits that overall improve the paper. That being said, I don't think the clarifications and edits address the major concerns:
>
> - The claims made in the abstract related to synthesis procedures and simulation workflows are still not properly supported. Saying that LLAMP performs better than vanilla LLMs is not enough to support that LLAMP can reliably perform those tasks and capabilities.
> - The machine learning innovation question is still not properly addressed and resolved. While putting together the APIs and performing the engineering work is valuable, this alone may be a better fit for a materials science publication. For the paper to be competitive a machine learning venue like ICLR comparing LLAMP with vanilla LLM prompting and flat planning is not enough given the diversity of RAG and tool calling methods that have been proposed. To make the paper stronger, the authors should: 1. Show that their proposed set of tools show improvement with multiple backend models consistently thereby showing the value of using the LLAMP set of tools for useful tasks. 2. Provide better grounding as to how their proposed method compares to other proposed RAG and tool calling methods. While a description of this should be there at minimum, ideally there would be experiments to provide empirical evidence.

---

> ### Author Response · Authors · 2024-11-24
>
> Thank you for the prompt feedback and valuable suggestions. We address our concerns below and have provided additional benchmarks to highlight the contributions of our work.
>
> **[Synthesis]** We have performed additional experiments to demonstrate the improvements of using LLaMP on inorganic materials synthesis. As there are multiple possible reaction pathways and synthesis conditions, there will be multiple valid synthesis recipes and make the benchmark open-ended. In light of this, we follow the positive-unlabeled (PU) synthesizability classification task proposed in [1] by randomly selecting positive (probable) and unlabeled (unlikely) synthesizable materials in their published dataset and compare the performances of LLaMP with different backend LLMs and baselines. LLaMP effectively enhances the performances of backbone GPT-4 and Sonnet LLMs by a significant margin of 20%. Please see table below
>
> | Model              | Accuracy | F1    | Precision    | Recall     |
> |--------------------|----------|-------|--------------|------------|
> | LLaMP (GPT-4)      | 0.800    | 0.773 | **0.895**    | 0.680      |
> | LLaMP (Sonnet)     | **0.818**| **0.812** | 0.848      | 0.780      |
> | GPT-4              | 0.600    | 0.649 | 0.578        | 0.740      |
> | Sonnet             | 0.530    | 0.230 | 0.636        | 0.140      |
> | Llama3             | 0.480    | 0.623 | 0.489        | **0.860**  |
> | Gemini             | 0.590    | 0.388 | 0.765        | 0.260      |
>
> **[Agentic simulation]** We have also run **50 molecular dynamics (MD) simulations** on randomly selected supercells of MP compounds. Our results show that LLaMP can successfully trigger the MD workflow for 96% of the cases, with 62% finishing 0.1 picosecond trajectories and 34% too slow to finish within timeout. The remaining 4% encounter some other unspecified situations (UNKNOWN). The further investigation indicates that for some materials LLaMP asks user for approval on the precise chemical formula to fetch the structure from MP.
>
>
> > The machine learning innovation question is still not properly addressed and resolved.
>
> **[Innovation]**  In the original submission, we have shown superior performance of LLaMP compared to specialized fine-tuned model Darwin [2], prompting method StructChem [3], and the basic RAG/tool method GPT-4+SerpAPI. We now appreciate the concurrent work of HoneyComb as a potential strong baseline RAG method. However, their proposed knowledge base is not interfaced with expert-curated, up-to-date computational materials databases esp. Materials Project database, and they (and most if not all the related works) focus on general capabilities such as question answering on MaScQA [4]. In this work, we however emphasize the importance of grounding LLMs on high-fidelity, real-time MP database for materials R&D, and we have shown the unprecedented precision performance of LLaMP to advance in this regard, which none of the concurrent works has adequately addressed, and many of them (including HoneyComb to our best knowledge) haven’t released reliable working codebases to support their experiments. Given the above and the fact that we are the first to implement the comprehensive API interface and quantify the grounding performance on Materials Project, the current available measures to compare LLaMP with other RAG frameworks are very much limited. Still, we propose a novel metric “self-consistency of response (SCoR)” to support our experiments and it shows that LLaMP is effective in our tasks that are practical, important, and meaningful in many aspects.
>
> Thank you again for your detailed review. We hope this clarifies the unique contributions of LLaMP and its relevance to both the ML and materials science communities. We have updated the manuscript to reflect the change, and we are happy to further clarify and provide additional supporting information.
>
>
> [1] Kim, Seongmin, et al. Large Language Models for Inorganic Synthesis Predictions, 2024
>
> [2] Xie, Tong, et al. DARWIN Series: Domain Specific Large Language Models for Natural Science, 2023
>
> [3] Ouyang, Siru, et al. Structured Chemistry Reasoning with Large Language Models, 2024
>
> [4] Zaki, M., & Krishnan, N. M. (2023). MaScQA: A Question Answering Dataset for Investigating Materials Science Knowledge of Large Language Models. arXiv preprint arXiv:2308.09115.

---

> ### Author Response · Authors · 2024-11-25
>
> **[Reliability with different backend models]**
>
> >  To make the paper stronger, the authors should: 1. Show that their proposed set of tools show improvement with multiple backend models consistently…
>
> We appreciate your valuable suggestion and we here provide additional experiments on different models. We hope our revisions clarify your concerns and strengthen our work.
>
> In the original submission, we conducted an ablation study (Table 4 in original submission, Table 5 in the updated submission) on the Bulk Modulus and Formation Energy tasks. We have now redone the benchmark of material properties prediction (Table 1) with different backend LLMs — GPT-4, Sonnet, Gemini, Llama3. We have updated the paper and provided the results below. The benchmark shows that LLaMP works well with GPT-4 and Sonnet as backend LLMs. As we stated in the original submission (line 320):
>
> > “we found LLaMP’s grounding performance correlates with the function-calling capability of backbone LLM: Claude-3.5-Sonnet (#1) > Gemini-1.5-Flash (#24) > and Llama3-8B (#46)’’
>
> Our framework can perform well on different backend LLMs provided that their function calling qualities are good enough to understand API schema reliably and stably generate relevant and correct input arguments. Our experiment shows that LLaMP improves different base models consistently, with GPT-4 and Sonnet more effective than others.
>
>
> *(3a) Bulk Modulus \( K \) (GPa)*
> | Model           | Precision ↓  | CoP ↑   | Confidence   | SCoR ↑  | MAE ↓   |
> |------------------|---------|---------|--------|---------|---------|
> | LLaMP (GPT-4)   | 2.698   | 0.900   | 1.000  | 0.900   | 14.574  |
> | LLaMP (Sonnet)  | 1.816   | 0.562   | 1.000  | 0.562   | 15.104  |
> | LLaMP (Gemini)  | 5.178   | 0.053   | 1.000  | 0.053   | 16.251  |
> | LLaMP (Llama3)  | 12.993  | 0.036   | 0.800  | 0.029   | 50.308  |
> | Sonnet          | 0.009   | 0.992   | 1.000  | 0.992   | 41.033  |
> | GPT-4           | 0.186   | 0.910   | 1.000  | 0.910   | 41.225  |
> | Gemini-Pro      | 6.065   | 0.169   | 1.000  | 0.169   | 43.429  |
> | Llama 3         | 11.222  | 0.010   | 1.000  | 0.010   | 41.874  |
>
> *(3b) Formation Energy $ \Delta H_f$ (eV)*
> | Model           | Precision ↓  | CoP ↑   | Confidence   | SCoR ↑  | MAE ↓   |
> |------------------|---------|---------|--------|---------|---------|
> | LLaMP (GPT-4)   | 0.006   | 0.994   | 0.940  | 0.934   | 0.007   |
> | LLaMP (Sonnet)  | 0.000   | 1.000   | 1.000  | 1.000   | 0.000   |
> | LLaMP (Gemini)  | 0.076   | 0.932   | 0.620  | 0.576   | 0.166   |
> | LLaMP (Llama3)  | 0.000   | 1.000   | 0.250  | 0.250   | 1.377   |
> | Sonnet          | 0.022   | 0.979   | 1.000  | 0.979   | 294.360 |
> | GPT-4           | 0.000   | 1.000   | 0.180  | 0.200   | 1.680   |
> | Gemini-Pro      | 0.467   | 0.657   | 1.000  | 0.657   | 1.412   |
> | Llama 3         | 2.346   | 0.139   | 0.960  | 0.137   | 4.657   |
>
>
> *(3c) Band Gap $E_g$ (eV)*
> | Model           | Precision ↓  | CoP ↑   | Confidence   | SCoR ↑  | MAE ↓   |
> |------------------|---------|---------|--------|---------|---------|
> | LLaMP (GPT-4)   | 0.000   | 1.000   | 0.800  | 0.800   | 0.000   |
> | LLaMP (Sonnet)  | 0.145   | 0.870   | 0.600  | 0.522   | 0.298   |
> | LLaMP (Gemini)  | 0.627   | 0.571   | 0.600  | 0.343   | 1.327   |
> | LLaMP (Llama3)  | 0.051   | 0.952   | 0.800  | 0.761   | 1.038   |
> | Sonnet          | 0.000   | 1.000   | 1.000  | 1.000   | 0.938   |
> | GPT-4           | 0.032   | 0.970   | 1.000  | 0.970   | 0.959   |
> | Gemini-Pro      | 0.034   | 0.968   | 1.000  | 0.968   | 0.994   |
> | Llama 3         | 0.051   | 0.954   | 1.000  | 0.954   | 1.039   |
>
> *(3d) Band Gap (Multi-element)*
> | Model           | Precision ↓  | CoP ↑   | Confidence   | SCoR ↑  | MAE ↓   |
> |------------------|---------|---------|--------|---------|---------|
> | LLaMP (GPT-4)   | 0.047   | 0.958   | 0.960  | 0.918   | 0.167   |
> | LLaMP (Sonnet)  | 0.046   | 0.962   | 1.000  | 0.962   | 0.304   |
> | LLaMP (Gemini)  | 0.003   | 0.997   | 0.500  | 0.997   | 0.637   |
> | LLaMP (Llama3)  | 0.169   | 0.848   | 0.800  | 0.678   | 1.094   |
> | Sonnet          | 0.000   | 1.000   | 0.500  | 1.000   | 0.644   |
> | GPT-4           | NaN     | NaN     | 0.000  | 0.000   | NaN     |
> | Gemini-Pro      | 0.168   | 0.849   | 0.600  | 0.509   | 0.989   |
> | Llama 3         | 0.181   | 0.837   | 0.920  | 0.770   | 1.051   |

---

### Official Review · Reviewer_PR3M · 2024-11-03

**Soundness:** 3
**Presentation:** 3
**Contribution:** 3
**Rating:** 8
**Confidence:** 4

**Summary:**

The authors propose LLaMP, a multimodal retrieval-augmented generation (RAG) framework leveraging hierarchical reasoning-and-acting (ReAct) agents to dynamically interact with Materials Project (MP), arXiv, Wikipedia, and atomistic simulation tools. To reduce the hallucination of Large Language Models (LLMs), the framework provides LLMs with high-fidelity material informatics derived from various sources. Consequently, LLaMP leverages hierarchical planning and correctly retrieves higher-order materials data and also combines different modalities to perform complex, knowledge-intensive inferences and operations essential for real-world research applications. LLaMP manifests enhanced performance in predicting key material properties, real-world applications in materials science, and language-driven simulations.

**Strengths:**

1. The paper presents a well-designed hierarchical ReAct agentic framework that dynamically manages multiple agents, each specialized in distinct tasks. This structured approach is highly modular, improving the model's accuracy and efficiency in handling complex workflows.

1. The authors provide a comprehensive discussion of LLaMP's strengths and limitations, offering valuable insights into its potential and constraints in materials science. This balanced analysis enhances the paper's utility for related researchers by clarifying LLaMP's practical applicability and areas for further development.

1. The self-correcting agent mechanism enables real-time error correction during tool usage and API interactions. This design minimizes the propagation of errors and enhances reliability in complex tasks.

1. The paper not only presents benchmarking results and analysis but also discusses the real-world applications in materials science. This applied focus, combined with LLaMP's use of high-fidelity data sources, makes it highly relevant and impactful for researchers.

**Weaknesses:**

1. The framework relies heavily on data from the Materials Project, which may restrict its application to new or less-explored materials. Although MP is comprehensive, the paper acknowledges that the database's coverage is not exhaustive, particularly for certain magnetic and bandgap configurations.

1. While the paper demonstrates LLaMP’s applications in materials science, the analysis is primarily based on one or two examples for each application. A more comprehensive experiment or in-depth discussion of success and failure cases should be helpful for exploring real-world applications.

1. The statistics for precision and CoP in Table 1 seem not consistent with the equation in Section 4.2. See the questions for details.

**Questions:**

1. According to Section 4.2, $CoP = exp(−Precision)$. However, a lot of Precision and CoP results in Table 1 do not match this relationship. For example, The Precision of LLaMP on bulk modulus is $2.698$, then $exp(-2.698) = 0.067$, but the reported value is $0.900$. Can the authors clarify that?

1. Have the authors tried LangGraph instead of LangChain for a more structured agentic workflow instead of a linear one? Langgraph will probably be beneficial as it may enhance the hierarchical planning in LLaMP, potentially improving agent coordination and task prioritization, especially for complex, multi-step processes.

1. In the example in Figure A.1, it seems that the supervisor gives mp-9258 as the answer to the query even though the two assistant agents never return any of the relevant information. So can I interpret that the example ends up with a hallucination? Is there any method to prevent this, and why is there not an observation step to capture the mismatch between the requested and returned MP IDs and to require the refinement of the action?

1. While LLaMP shows strength in typical systems (e.g., Si-O, Li-based compounds), how does it perform on less common or highly complex systems, such as multinary oxides or intermetallic compounds?

---

> ### Author Response · Authors · 2024-11-15
>
> Thank you for your review. We appreciate the opportunity to clarify key design choices and address your comments.
>
> **W1:**
>
> We would like to clarify that LLaMP is intended to ground LLMs on the current, high-quality, and up-to-date data in the MP database and it is supposed to refuse to respond when the factual information of unknown materials is not available from equipped tools including MP API, SerpAPI, etc. This behavior is desirable and aligns with the common practices in real materials discovery campaigns as we would like the agent to admit its limitations and take necessary actions before conducting further studies, for example running extra simulations. Our approach here, instead of imposing limitations, provides an essential step for full-stack approaches from data exploration to autonomous in-silico experiments for materials R&D.
>
> While we acknowledge MP’s coverage is not exhaustive, MP offers one of the most comprehensive and high-fidelity information, and is continuously updated and vetted by the materials science community. In contrast to general web-based text corpora that may contain outdated or contextually misaligned information, MP’s curated dataset minimizes such risks, enhancing reliability for downstream applications where consistency and scientific validity are critical.
>
>
> **W2:**
>
> In the paper, we present magnetism benchmarks at scale for LLaMP and vanilla GPT-4 and GPT-3.5 (800 materials randomly selected from all unary, binary, and ternary compounds in MP). We would also like to highlight that the suitable benchmark dataset for testing RAG performance of LLM in science is sparse and only available concurrently (https://arxiv.org/abs/2411.00177). Given this context, we curate the new dataset and aim to benchmark the quality of grounding in the MP database.
>
> We also prioritized representative examples across different applications to demonstrate LLaMP’s versatility in handling complex real-world tasks. To supplement these, in the revision we provide additional cases in Supplementary A.2 where LLaMP admits ignorance or explicitly states that the response is based on intrinsic knowledge.
>
> **W3, Q1:**
>
> Table 1 shows the material-wise average values across five runs. To be more precise, each material has its own precision, CoP, confidence values and SCoR is calculated separately across five runs. The SCoR were then averaged over different materials. A detailed example metric calculation will be provided in appendix A.2. We also provide the code in the submission supplementary and include the detailed metric calculation.
>
> **Q2:**
>
> We have been aware of LangGraph’s development but have not yet implemented it, as our project began before LangGraph was available. We believe that incorporating LangGraph could enhance flexibility and enable more dynamic routing between assistant agents, and we leave it to explore in future work.
>
> **Q3:**
> The full response is indeed included in the API output; however, due to its length, it was not fully visualized in the paper. We attached the full example notebook in submission supplementary material along with the code.
>
> **Q4:**
> Thank you for the suggestion. Although MP is one of the most authoritative material databases, it is heavily skewed to oxides, due in part to the abundance and complexity of oxides. In fact, in Figure 2 there is one multinary oxide Eu2B5BrO9 and LLaMP effectively grounds the LLM to the MP value. We recognize that MP has fewer intermetallic compounds than other databases such as AFLOW. We have added this point to the limitation section. Thank you.

---

> > ### Comment · Reviewer_PR3M · 2024-11-17
> >
> > Thanks for the response. The authors' response has addressed most of my concerns. With that, I will raise my score.

---

> ### Author Response · Authors · 2024-11-19
>
> Thank you for reviewing our clarifications. We appreciate your precious time and constructive suggestion.

---

### Official Review · Reviewer_ziiq · 2024-11-04

**Soundness:** 3
**Presentation:** 3
**Contribution:** 1
**Rating:** 3
**Confidence:** 3

**Summary:**

The manuscript describes the introduction and test of LLaMP, an LLM fine tuned to interact with the materials science data from the Materials Project database and run atomistic simulations via a workflow interface.

**Strengths:**

The introduced LLM is very likely of great interest for the materials science community, in particular because it is ready to be used and it is fine tuned on one of the largest and most consistent corpus of materials-science data, especially computational (atomistic-simulation) data.
Results suggest it would be a useful tool used for materials-science research.

**Weaknesses:**

There is no methodological advance worth highlighting here. The described "hierarchical agent framework" is a simple device to design and implement, while the metric assessing consistency is reasonable for reporting and discussing results but does not strike as a major advance.
In view of these strengths and weaknesses, the paper seems more indicated for a materials-science specialized journal such as npj Computational Materials.

**Questions:**

- Figure 2, which reports with Table 1 the main results of the manuscript, is barely readable.
Besides working harder on contrasting colors overall visibility, the authors should explain all symbols' choices and consider either reducing the amount of data shown or make the figure in more panels (possibly reporting some crucial example in the main text and the rest in an appendix).
- in Table 1, confidence scores equal exactly to 1 (at least to the third digit precision), especially if combined with MAE equal to 0.000 (see electronic band gaps of common elements for LLaMP) look unlikely good and suggest the prediction use training data. This aspect would need discussion in presenting the results, and probably a different design of the performance tests.

---

> ### Author Response · Authors · 2024-11-13
>
> We appreciate the reviewer's time and comment. Our methodological advance is the implementation of large-scale API orchestration and hierarchical agent itself, which resolves the challenges when grounding LLM on one of the largest and the most coherent source of materials database. There are many MP API endpoints having more than ten arguments https://api.materialsproject.org/docs, and the logic is highly challenging to a layman.
>
> We stress that implementing such a large and reliable tool is far beyond trivial and important for reproducibility and pointing the field toward the right path, which many previous and concurrent work do not fulfill. This work relies on retrieval-augmented generation instead of fine-tuning/training on the MP database, and we prove that such a method is necessary to generate high fidelity/quality response that both scientists and ML practitioners should care about for the materials research pipeline. We believe that ICLR is the right stage for pointing the community to clearly see this purpose and benefiting the future work, e.g. fine tuning LLM on MP database of diverse chemistry and multimodal data (tensors, 3D crystals).
>
> There is no training in our framework. LLaMP performs RAG on API response (so called in-context grounding/learning). We did a manual API request as our MP ground truth data, and LLaMP performs API requests autonomously without human intervention. For the bandgap of common materials, LLaMP successfully retrieves the correct data for all materials over 5 trials and hence the high confidence and low MAE, thanks to the coherence of MP database and API requests. We could see that since LLM generation is still probabilistic, LLaMP for less common cases sometimes fetches other polymorphic crystals (still valid materials satisfying the description in the query, e.g. valid chemical formula). This induces variance seen in other properties in Table 1 when such an ambiguity is always present if the query is not precise enough. This is an opportunity and limitation simultaneously, as it offers a great flexibility in user query to access diverse data on MP while if the user is not descriptive and vigilant in their query, LLaMP might request different valid data from time to time. With that said, we are to the best of our knowledge the first to quantify such variance and demonstrate that LLaMP effectively improves this.
>
> Thank you again for your comment. We have added more text and self-explanatory example and revised the limitation section accordingly to address the point mentioned above. We will upload the updated manuscript all together with other revisions shortly.

---

> ### Author Response · Authors · 2024-11-25
>
> We would like to thank you again for the comments and follow up if we could provide more clarification. We have provided additional experiments and updated the manuscript that also addresses the related concerns by other reviewers. We hope our revisions clarify the unique contributions of LLaMP and its relevance to both the ML and materials science communities.
>
> **AQ1** We have adjusted the presentation of Figure 2 for better visualization of the boxplots. We have also added the annotation to the captions
>
> **AQ2** We have thoroughly performed all of the Table 1 benchmarks once again and have provided the results for additional backend LLMs (Sonnet, Gemini, Llama3). We still saw the perfect grounding of LLaMP (GPT-4) on bandgap with 0 MAE despite decreased SCoR to 0.8. Our framework can perform well on different backend LLMs provided that their function calling qualities are good enough to understand API schema reliably and stably generate relevant and correct input arguments. Our experiment shows that LLaMP improves different base models consistently, with GPT-4 and Sonnet more effective than others.
>
> *(3a) Bulk Modulus \( K \) (GPa)*
> | Model           | Precision ↓  | CoP ↑   | Confidence   | SCoR ↑  | MAE ↓   |
> |------------------|---------|---------|--------|---------|---------|
> | LLaMP (GPT-4)   | 2.698   | 0.900   | 1.000  | 0.900   | 14.574  |
> | LLaMP (Sonnet)  | 1.816   | 0.562   | 1.000  | 0.562   | 15.104  |
> | LLaMP (Gemini)  | 5.178   | 0.053   | 1.000  | 0.053   | 16.251  |
> | LLaMP (Llama3)  | 12.993  | 0.036   | 0.800  | 0.029   | 50.308  |
> | Sonnet          | 0.009   | 0.992   | 1.000  | 0.992   | 41.033  |
> | GPT-4           | 0.186   | 0.910   | 1.000  | 0.910   | 41.225  |
> | Gemini-Pro      | 6.065   | 0.169   | 1.000  | 0.169   | 43.429  |
> | Llama 3         | 11.222  | 0.010   | 1.000  | 0.010   | 41.874  |
>
> *(3b) Formation Energy $ \Delta H_f$ (eV)*
> | Model           | Precision ↓  | CoP ↑   | Confidence   | SCoR ↑  | MAE ↓   |
> |------------------|---------|---------|--------|---------|---------|
> | LLaMP (GPT-4)   | 0.006   | 0.994   | 0.940  | 0.934   | 0.007   |
> | LLaMP (Sonnet)  | 0.000   | 1.000   | 1.000  | 1.000   | 0.000   |
> | LLaMP (Gemini)  | 0.076   | 0.932   | 0.620  | 0.576   | 0.166   |
> | LLaMP (Llama3)  | 0.000   | 1.000   | 0.250  | 0.250   | 1.377   |
> | Sonnet          | 0.022   | 0.979   | 1.000  | 0.979   | 294.360 |
> | GPT-4           | 0.000   | 1.000   | 0.180  | 0.200   | 1.680   |
> | Gemini-Pro      | 0.467   | 0.657   | 1.000  | 0.657   | 1.412   |
> | Llama 3         | 2.346   | 0.139   | 0.960  | 0.137   | 4.657   |
>
>
> *(3c) Band Gap $E_g$ (eV)*
> | Model           | Precision ↓  | CoP ↑   | Confidence   | SCoR ↑  | MAE ↓   |
> |------------------|---------|---------|--------|---------|---------|
> | LLaMP (GPT-4)   | 0.000   | 1.000   | 0.800  | 0.800   | 0.000   |
> | LLaMP (Sonnet)  | 0.145   | 0.870   | 0.600  | 0.522   | 0.298   |
> | LLaMP (Gemini)  | 0.627   | 0.571   | 0.600  | 0.343   | 1.327   |
> | LLaMP (Llama3)  | 0.051   | 0.952   | 0.800  | 0.761   | 1.038   |
> | Sonnet          | 0.000   | 1.000   | 1.000  | 1.000   | 0.938   |
> | GPT-4           | 0.032   | 0.970   | 1.000  | 0.970   | 0.959   |
> | Gemini-Pro      | 0.034   | 0.968   | 1.000  | 0.968   | 0.994   |
> | Llama 3         | 0.051   | 0.954   | 1.000  | 0.954   | 1.039   |
>
> *(3d) Band Gap (Multi-element)*
> | Model           | Precision ↓  | CoP ↑   | Confidence   | SCoR ↑  | MAE ↓   |
> |------------------|---------|---------|--------|---------|---------|
> | LLaMP (GPT-4)   | 0.047   | 0.958   | 0.960  | 0.918   | 0.167   |
> | LLaMP (Sonnet)  | 0.046   | 0.962   | 1.000  | 0.962   | 0.304   |
> | LLaMP (Gemini)  | 0.003   | 0.997   | 0.500  | 0.997   | 0.637   |
> | LLaMP (Llama3)  | 0.169   | 0.848   | 0.800  | 0.678   | 1.094   |
> | Sonnet          | 0.000   | 1.000   | 0.500  | 1.000   | 0.644   |
> | GPT-4           | NaN     | NaN     | 0.000  | 0.000   | NaN     |
> | Gemini-Pro      | 0.168   | 0.849   | 0.600  | 0.509   | 0.989   |
> | Llama 3         | 0.181   | 0.837   | 0.920  | 0.770   | 1.051   |

---

### Author Response · Authors · 2024-12-02

We sincerely thank the reviewers for their valuable feedback and constructive comments. We have carefully considered all points raised and have made revisions to address the concerns. Here, we reiterate the concerns and the contribution of the work.
1. **Proposed a hierarchical+ReAct agentic framework that enhances the API calling reliability and effectively ground the LLMs on high-fidelity material informatics:**
   We propose a hierarchical+ReAct agentic framework that can stably ground material information, enhancing the reliability of material data processing and retrieval. Many current LLM material query and discovery methods suffer from hallucination and premature stopping, and are still not aware of up-to-date data, limiting their practical usage in real-world application settings. LLaMP framework permits the chance to improve overtime along with both LLM capabilities and new data on Materials Project, and gracefully and faithfully bail out if the high-fidelity data are not available. This is important for downstream scientific applications. In Appendix A4, we show how our framework acknowledges its limitations and refuses to respond to ambiguous queries or if the relevant data is unavailable.

2. **Extensive experiments on real-world downstream applications:**
   To demonstrate how our model can be useful and incorporated into a real-world lab setting, we study its capacity to provide synthesis recipes, perform molecular simulations, edit crystal structures, and retrieve material informatics. We acknowledge the point raised by reviewer X8Ga regarding the scale of our experiments. Therefore, we have conducted additional experiments on synthesis recipes and molecular simulations. Please refer to our discussion with reviewer X8Ga for the experimental details.
   - **Inorganic Materials Synthesis:**
     Following the positive-unlabeled (PU) synthesizability classification task proposed by Kim et al. [1], our results show that LLaMP enhances the performance of backbone GPT-4 and Sonnet LLMs by a significant margin of 20%.
   - **Molecular Dynamics Simulations:**
     We performed 50 simulations on randomly selected supercells from the MP database. LLaMP successfully initiated the MD workflow for 96% of the cases, demonstrating robustness in practical settings.

3. **The proposed framework is robust across all model backends and state-of-the-art models:**
   We have conducted extensive comparisons of our framework with state-of-the-art models such as StructChem [3] and Darwin [2], as well as baseline retrieval-augmented generation (RAG) methods like GPT-4 with SerpAPI. The ablation study (see Table 4 in the appendix) further shows that our framework consistently improves performance across multiple backend models given a strong enough model, including sonnet and GPT-4. Our results demonstrate that our framework not only surpasses these models in performance but does so consistently across various backend models. Please refer to our discussion with reviewer X8Ga for the more details.

We appreciate the reviewers’ time and thoughtful feedback. We kindly ask if our responses have adequately addressed your concerns. Please let us know if there are any remaining points of clarification needed.

[1] Kim, Seongmin, et al. Large Language Models for Inorganic Synthesis Predictions, 2024
[2] Xie, Tong, et al. DARWIN Series: Domain Specific Large Language Models for Natural Science, 2023
[3] Ouyang, Siru, et al. Structured Chemistry Reasoning with Large Language Models, 2024

---

### Meta-Review · Area_Chair_au6M · 2024-12-22

**Metareview:**

In this work, authors present a multimodal retrieval-augmented generation (RAG) framework that uses hierarchical reasoning-and-acting (ReAct) agents to interact with Materials Project database and run atomistic simulations. The paper claims improved accuracy in material property prediction, reduced hallucination through data retrieval, and capabilities in synthesis recommendations and molecular simulations. The framework shows lower MAE and better self-consistency across multiple LLM backends for various materials science tasks. The work's strengths lie in its practical utility, comprehensive evaluation, novel consistency metrics, and strong empirical results, supported by reproducible implementation. However, significant weaknesses undermine its suitability for ICLR. The core methodology primarily combines existing approaches without fundamental theoretical advances in LLM or agent architectures. The heavy dependence on Materials Project database raises concerns about generalizability, while claims about synthesis capabilities lack comprehensive validation. The paper's emphasis on engineering implementation over theoretical innovation, coupled with inadequate comparisons to state-of-the-art RAG/tool-use methods, suggests it would be better suited for a materials science venue. While LLaMP represents a valuable contribution to materials science research tooling, its limited ML methodology advancement and focus on practical implementation over theoretical insights do not meet the standards expected for ICLR. The authors should consider expanding the theoretical framework, broadening experimental validation, and resubmitting to a domain-specific journal where the practical impact would be more appropriately valued.

**Additional Comments On Reviewer Discussion:**

During the rebuttal period, several key concerns were raised and addressed across multiple rounds of discussion.

Reviewer X8Ga highlighted two fundamental issues: insufficient ML innovation and inadequate experimental validation. While the authors defended their contribution as a practical framework managing complex API orchestration, and provided additional experiments showing 20% improvement in synthesizability classification and 96% success rate in molecular dynamics simulations, these additions primarily reinforced the engineering nature of the work rather than establishing theoretical innovation. The authors also expanded their benchmarks across multiple LLM backends (GPT-4, Sonnet, Gemini, Llama3) to demonstrate framework robustness.

Reviewer PR3M focused on technical implementation details, questioning the distinction between agents and framework limitations. The authors clarified the roles and specializations of different agents, explaining how the supervisor agent coordinates task-specific assistant agents. Meanwhile, Reviewer ziiq requested stronger evidence for synthesis and simulation claims, which the authors addressed through their additional experiments.

Reviewer PUgh raised specific concerns about metrics and visualization clarity, particularly regarding the SCoR metric and figure readability. The authors responded by clarifying their precision metric definition, improving figure visualization, and releasing code for reproducibility. This technical dialogue led to concrete improvements in the paper's presentation and reproducibility.

While authors' did several additional experiments to address the concerns and strengthened the paper's empirical validation, the fundamental issue of limited theoretical contribution to ML methodology remained unresolved. The additional experiments and clarifications, though valuable, ultimately reinforced the work's identity as an engineering achievement rather than a ML research advancement. Considering these points, it is recommended that the paper would be better suited for a materials science venue where its applied engineering contribution would be more appropriately valued.

---

### Decision · Program_Chairs · 2025-01-22

Reject